# Dual Data Alignment Makes AI-Generated Image Detector Easier Generalizable

**Ruoxin Chen[1], Junwei Xi[2], Zhiyuan Yan[3], Keyue Zhang[1], Shuang Wu[1],**
**Jingyi Xie[4], Xu Chen[2], Lei Xu[5], Isabel Guan[6†], Taiping Yao[1†], Shouhong Ding[1]**
[1]Tencent YouTu Lab, [2]East China University of Science and Technology,
[3]Peking University, [4]Renmin University of China,
[5]Shenzhen University, [6]Hong Kong University of Science and Technology
[†] Corresponding Authors
{cusmochen, taipingyao}@tencent.com, eeguan@ust.hk

## Abstract

The rapid increase in AI-generated images (AIGIs) underscores the need for detection methods. Existing detectors are often trained on biased datasets, leading to overfitting on spurious correlations between non-causal image attributes and real/synthetic labels. While these biased features enhance performance on the training data, they result in substantial performance degradation when tested on unbiased datasets. A common solution is to perform data alignment through generative reconstruction, matching the content between real and synthetic images. However, we find that pixel-level alignment alone is inadequate, as the reconstructed images still suffer from frequency-level misalignment, perpetuating spurious correlations. To illustrate, we observe that reconstruction models restore the high-frequency details lost in real images, inadvertently creating a frequency-level misalignment, where synthetic images appear to have richer high-frequency content than real ones. This misalignment leads to models associating high-frequency features with synthetic labels, further reinforcing biased cues. To resolve this, we propose Dual Data Alignment (DDA), which aligns both the pixel and frequency domains. DDA generates synthetic images that closely resemble real ones by fusing real and synthetic image pairs in both domains, enhancing the detector's ability to identify forgeries without relying on biased features. Moreover, we introduce two new test sets: DDA-COCO, containing DDA-aligned synthetic images, and EvalGEN, featuring the latest generative models. Our extensive evaluations demonstrate that a detector trained exclusively on DDA-aligned MSCOCO improves across diverse benchmarks. Code is available at https://github.com/roy-ch/Dual-Data-Alignment.

## 1 Introduction

The rise of AIGIs [12, 18, 41, 49] poses risks to digital security, including the potential for misinformation, fraud, and copyright violations [12, 20, 21, 18, 34, 54, 52, 51, 47, 24, 36]. This severe security issue underscores the urgent need for reliable detection methods to differentiate synthetic images from authentic ones. Despite advances in AIGI detection techniques [4, 32, 35], the rapid evolution of generative models and the emergence of new architectures present cross-domain generalization challenges. This is especially evident in zero-shot scenarios involving previously unseen generation paradigms.

The generalizability of AIGI detectors is hindered by dataset biases [13, 4, 33, 14]. Existing datasets often exhibit systematic discrepancies in attributes unrelated to the authority. Works [38] illustrate semantic bias through word frequency analysis, and studies [35] demonstrate image size bias by

39th Conference on Neural Information Processing Systems (NeurIPS 2025).

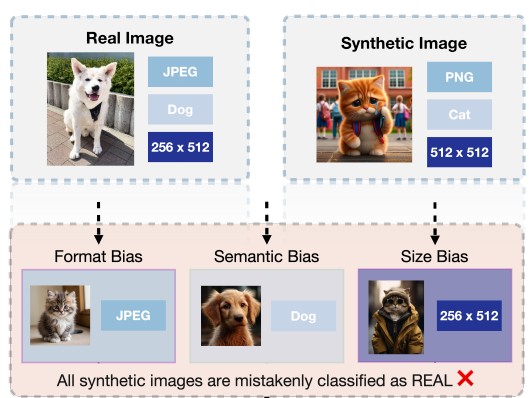

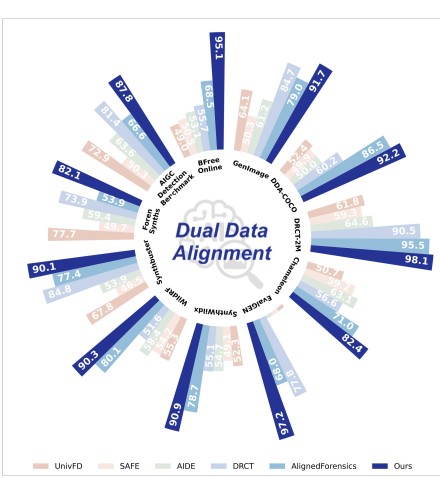

Figure 1: **Illustration of dataset bias.** Top row: Real/synthetic images show disparities in format, content, and size. Real images are typically in JPEG, with varying sizes and centered semantics. Bottom row: Detectors trained on datasets containing these discrepancies are prone to learning biased features, incorrectly associating authenticity with format, image size, or semantics.

Figure 2: **Overall comparison between detection methods on 11 benchmarks.** Our model is exclusively trained on DDA-aligned MSCOCO data. The consistent outperformance of DDA on 4 in-the-wild (Chameleon, WildRF, BFree-Online and SynthWildx) and 7 manually-crafted benchmarks validates the generalizability. Detailed results are provided in Section 4.

analyzing on the datasets where synthetic images are uniformly sized as multiples of $128 \times 128$. These non-causal features could be exploited by models to distinguish real from synthetic images, resulting in biased detector performance that fails to generalize across different datasets. Figure 1 visually illustrates such bias. **Dataset alignment holds promise in addressing the issue of dataset bias** by ensuring synthetic images closely resemble real ones, *excluding authenticity-related factors* and *directing detectors to focus on forgery-related cues*. Specifically, studies [13] reveal systematic discrepancies in format and size biases: real images are JPEG-encoded and vary in size, whereas synthetic images are uniformly PNG-encoded and fixed in size. SemGIR [53], DRCT [4], B-Free [14] aim to mitigate content discrepancies using diffusion reconstruction techniques that generate images semantically similar to real ones. Works [15, 55, 58] prevent models from learning semantics-dependent features by breaking images into patches and shuffling them.

However, in this paper, we ask: *Does reconstruction truly eliminate potential misalignment and bias?* Our answer is **no**. We find that although reconstruction-based methods align datasets at the pixel level, they still **introduce subtle misalignments at the frequency level**. Specifically, generative reconstruction based data alignment tends to preserve or even amplify details across all frequency bands. In particular, reconstructed images often restore high-frequency components that are diminished in real images—typically due to compression during transmission or storage, where such components are removed to reduce file size because they have little impact on human visual perception. Consequently, **synthetic images exhibit disproportionately strong high-frequency details, whereas real images contain much weaker ones**, creating a noticeable discrepancy in the magnitude of high-frequency components rather than in their semantic content. This spurious correlation can lead detectors to overfit these frequency cues, mistakenly identifying high-frequency richness as an indicator of synthetic origin.

In this paper, we propose **D**ual **D**ata **A**lignment (DDA), an effective technique that aligns synthetic images with real ones across both pixel and frequency domains. DDA consists of three steps: 1) VAE reconstruction for pixel alignment, 2) high-frequency fusion to eliminate bias, and 3) pixel mixup for further alignment in the pixel domain. As shown in Figure 2, **a single model trained on DDA-aligned MSCOCO demonstrates significant improvements across benchmarks: +11.4% on Chameleon, + 26.6% on BFree-Online and + 19.4% on EvalGEN, with much lower fluctuations across subsets – usually 1/2 that of baselines.** We attribute this significant performance boost to the data alignment process: when synthetic images are carefully aligned with real images across key domains, the model learns a tighter, more transferable decision boundary, enhancing generalizability

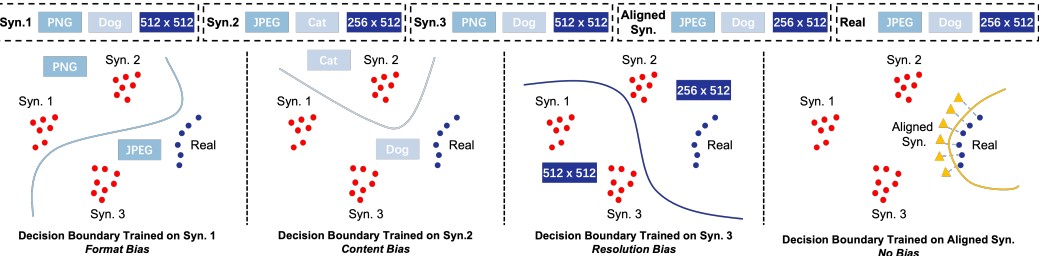

Figure 3: Visual illustration of how dataset bias affects decision boundaries. **Left three panels:** Detectors trained on biased data—where synthetic images (e.g., Syn.1–3) differ from real images in format, content, or resolution—tend to learn spurious decision boundaries. **Right:** When synthetic images are carefully aligned with real images across multiple aspects, the model can learn a tighter decision boundary that more accurately encompasses the real data.

to unseen data, as demonstrated in Figure 3. Moreover, we introduce two new evaluation datasets: 1) DDA-COCO, a test set consisting of real images from MSCOCO and their DDA-aligned counterparts. This dataset evaluates whether the detector captures inherent discriminative features or relies on other biases. Prior detectors suffer significant performance drops on DDA-COCO. 2) EvalGEN, a test set consisting of FLUX, GoT, Infinity, NOVA, and OmniGen, which includes both advanced auto-regressive and diffusion generators, serving for measuring detectors' generalizability under newly evolved generative models.

## 2   Related Works

**AIGI Detection.**   CNNSpot [42] trains a vanilla CNN model, finding that detectors easily recognize synthetic images from seen models but struggle to generalize to unseen ones. UnivFD [32] employs CLIP as backbone, showing the improvements in generalizability in detecting unseen generators. Subsequent works [25, 38, 55, 50] explore model architectures and image preprocessing for more generalizable detection. C2P-CLIP enhances the pretrained CLIP backbone for AIGI detection by injecting 'real' and 'fake' concepts. Works [39, 6, 23, 19, 57] exploit frequency domain artifacts, showing that frequency artifacts could well discriminate. NPR [40] explores the upsampling artifact in generative models. Vision–language approaches [27, 17, 26, 5, 46, 45] pursue explainable detection by leveraging VLMs' semantic priors. However, these methods' generalizability is limited by either content bias or frequency-level bias, with a chance of exploiting non-causal features like image format, which can degrade performance on unbiased test sets.

**Dataset alignment.**   The evaluation bias issue in AIGI detection is firstly introduced in the work [13], showing that image format and size are common biases unintentionally exploited by detectors. FakeInversion [3] introduces a bias-reduced evaluation benchmark, mitigating thematic and stylistic biases by collecting synthetic images that match real images in both content and style. A line of subsequent works explores eliminating bias in the training set to enhance generalizability. SemGIR [53] regenerates synthetic images by semantic-level reconstruction conditioned on the real counterpart's description, aiming to better align synthetic and real images semantically. DRCT [4] employs diffusion reconstruction for improved semantic alignment. B-Free [14] addresses dataset bias through self-conditioned inpainted reconstructions and content augmentation. However, this inpainting paradigm can alter the center object, corrupting the semantic alignment. AlignedForensics [33] performs simple VAE reconstruction without latent space manipulation, resulting in synthetic images that closely match real images in semantics and resolution. However, both B-Free and AlignedForensics overlook format alignment, creating space for JPEG-based shortcuts in discrimination.

## 3   Methodology

### 3.1   Motivation and Analysis

**Misaligned Dataset.**   In the absence of additional supervision, detectors rely exclusively on the training set to learn the concept of 'real' versus 'synthetic'. When these two classes' data differ systematically in non-causal attributes—such as compression format or semantic content—the model may incorrectly learn to associate these irrelevant features with authenticity. These spurious signals

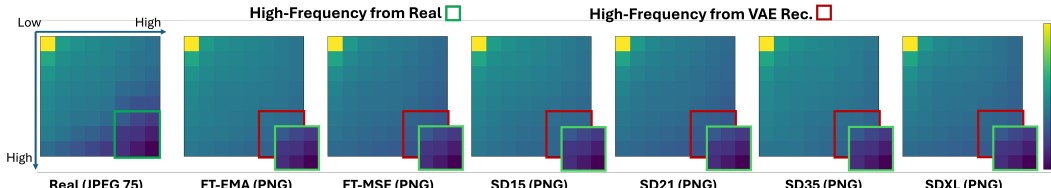

Figure 4: **Visualization of frequency domain energy using 2D DCT.** The left column shows a real image, while the remaining columns display images reconstructed by VAEs from various Stable Diffusion models. The grids represent frequency components, with the top-left and bottom-right indicating low- and high-frequency regions, respectively. Lighter areas correspond to higher energy. Real images in JPEG format exhibit darker high-frequency regions compared to VAE reconstructions, indicating weaker high-frequency content in real images.

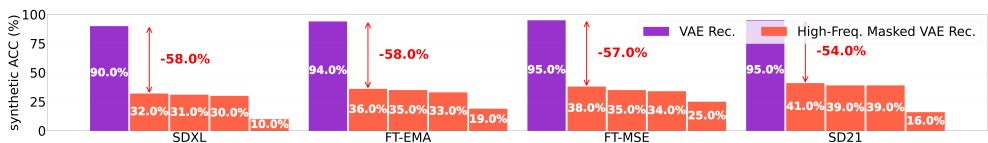

Figure 5: **Evidence for the existence of biased frequency-based features to discriminate reconstructed images.** We apply a binary mask to the DCT coefficients, systematically nullifying high-frequency components where either the horizontal or vertical frequencies exceed 95%, 90%, 85% and 80% of their respective spectral ranges to generate High-Freq. Masked VAE Rec.

are often more salient than subtle, genuine artifacts that actually distinguish real from synthetic images, making it more difficult for the model to learn truly generalizable features.

**Reconstruction-based Alignment.** To align synthetic images with real ones, some approaches [53, 59, 14] employ txt2img generative models to generate images with similar semantic content, conditioned on the image label or image captions obtained through pretrained models. However, images generated using this approach often differ from the originals due to the lack of strong and detailed supervision, which prevents the generated images from fully matching the original images in all semantic details. DRCT [4, 14] leverages Img2Img diffusion reconstruction, directly using the image itself to guide the reconstruction of a real image $x$ into a synthetic counterpart $\hat{x}$ as follows:

$$\hat{x} = \text{Decoder}(\hat{z}), \quad \text{where} \quad \hat{z} = z + \epsilon_t - \epsilon_\theta(z, t), \quad z = \text{Encoder}(x). \tag{1}$$

where $z$ represents the encoded latent of the real image, while $\hat{z}$ is modified by adding noise and subsequently denoising, creating new latents that subtly differ from $z$. However, such self-supervised diffusion reconstruction can still lead to changes in image details due to modifications in the latent space, which is responsible for the generation of semantics. The work [33] further simplifies the reconstruction process by using a Variational Autoencoder (VAE)—a submodule used in all stable diffusion generators—without any modification to the latent. This approach generates images that closely match the original real image at the pixel level.

$$\hat{x} = \text{Decoder}(z), \quad \text{where} \quad z = \text{Encoder}(x). \tag{2}$$

**Frequency-Level Misalignment Exists and Can Be Exploited.** Frequency domain has been widely explored in AIGI detectors [37, 23, 30, 48], demonstrating that frequency information is crucial for AIGI detection. This motivates us to revisit the frequency-domain alignment. Surprisingly, despite pixel-level alignment, synthetic counterparts exhibit significant discrepancies in high-frequency content. Figure 4 visualizes this discrepancy between the real image and synthetic images reconstructed using various VAEs. Real images are often with relatively poor high-frequency information, which is due to JPEG compression removing high-frequency details. Having identified this frequency-level discrepancy, another question arises: "Can this disparity be leveraged, or are we overestimating its impact?" To evaluate its effect, we assess the impact by measuring the variance in

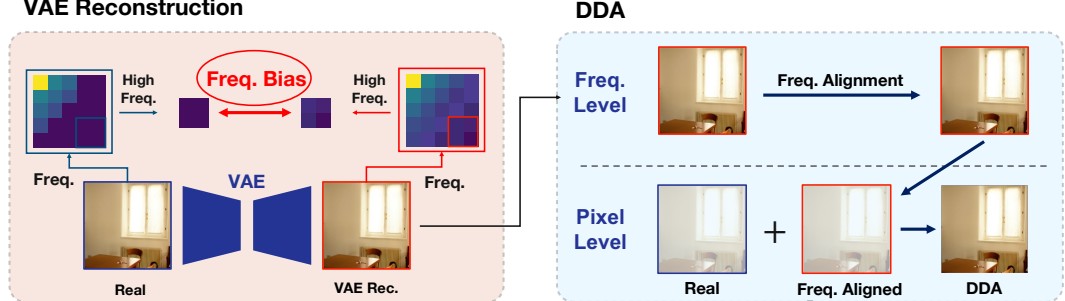

Figure 6: DDA pipeline. **Left:** VAE-reconstructed images differ from real ones in the intensity of high-frequency components. **Right:** DDA fuses high-frequency information from real images into the VAE-reconstructed images to align them in the frequency domain. Then, DDA uses pixel-level mixup of real and frequency-aligned images to further align them in the pixel domain.

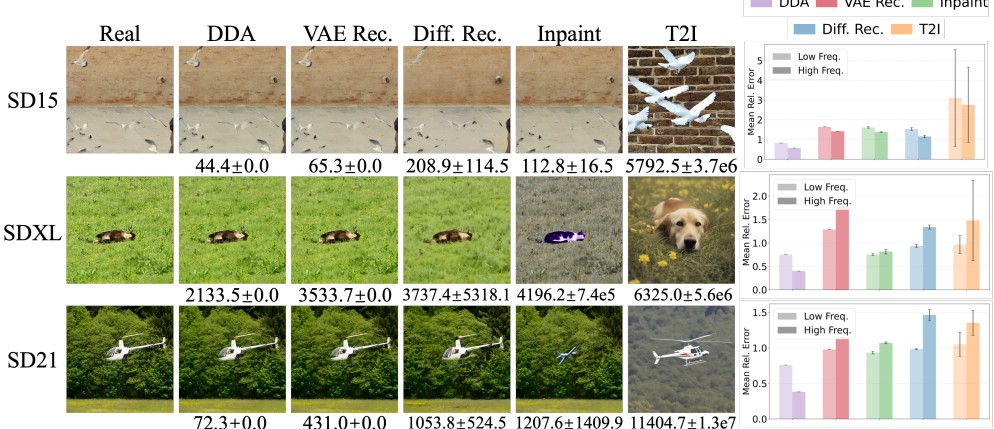

Figure 7: Comparison of various image processing methods based on loss with respect to the real image. **Left:** Comparison of image processing methods across three Stable Diffusion model series (SD15, SDXL, SD21) displaying real images alongside processed versions using DDA, VAE reconstruction (VAE Rec.), diffusion reconstruction (Diff. Rec.), masked inpainting with prompts (Inpaint), and text-to-image generation (T2I). Mean squared error (MSE) values relative to the real image are presented beneath each processed image, and each mse value is calculated by generating 100 images. **Right:** Visualization of relative error metrics for each processing method across the same model series, segregated into low frequency and high frequency bands as calculated using discrete Fourier transform (DFT). Bar charts illustrate comparative error magnitudes across different reconstruction techniques and frequency components. Both pixel-level and frequency-level analyses indicate that DDA produces synthetic images most similar to the real images.

detector performance on VAE-reconstructed images. As shown in Figure 5, the empirical results are striking: visually identical VAE-reconstructed images are detected by the frequency-based detector SAFE [23] with a 93% success rate, indicating a significant difference in the frequency domain. However, when we mask high-frequency information slightly, the detection rate drops dramatically. This substantial decline cannot be attributed solely to information loss; rather, it suggests that detectors exploit biased features—specifically, the richer high-frequency details in synthetic images due to their not undergoing JPEG compression, unlike real images.

## 3.2 Dual Data Alignment

Motivated by the previous observation, we propose DDA, a technique that generates synthetic images aligned with real ones in both the pixel and frequency domains to mitigate the learning of biased features. As illustrated in Figure 6, DDA consists of three steps: 1) **VAE Reconstruction:** Generate

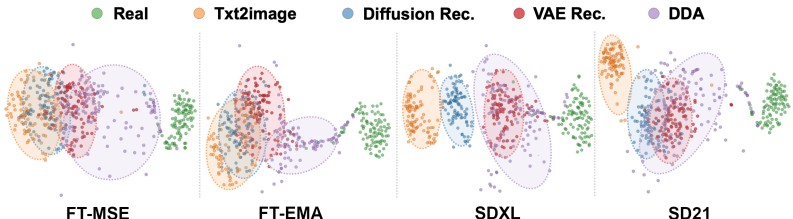

Figure 8: t-SNE visualizations comparing real and generated images, illustrating the proximity of synthetic image cluster centers to real images in feature space. The ordering of proximity—from closest to farthest—is: DDA, VAE reconstruction, diffusion reconstruction, and text-to-image (T2I) generation. These results indicate that DDA produces synthetic samples most closely aligned with real images near the data manifold boundary, thereby facilitating the learning of a tighter and more generalizable decision boundary.

pixel-wise similar images containing VAE-specific artifacts. 2) **Frequency-Level Alignment:** We identified that discrepancies in the frequency domain primarily arise from JPEG compression. To mitigate this, we align the frequency by applying the same JPEG compression with an equivalent quality factor to both real and VAE-reconstructed images. In practice, we estimate the quality factor of each real image before training and apply the same compression to its reconstructed counterpart during training. 3) **Pixel-Level Alignment:** Apply mixup between real and frequency-aligned images to ensure pixel-domain alignment. A closely aligned synthetic image is generated as follow:

$$x_{\mathrm{mix}} = r_{pixel} \cdot x_{real} + (1 - r_{pixel}) \cdot x_{syn}. \tag{3}$$

where $r_{pixel} \in [0, 1]$ controls the degree of pixel-level alignment. A higher $r_{pixel}$ value yields a closer synthetic image in the pixel space. In practice, $r_{pixel}$ is sampled from a uniform distribution $\mathcal{U}(0, R_{pixel})$. Together, these steps ensure that the resulting synthetic images preserve generative artifacts while maintaining close alignment with real data, both spectrally and spatially.

The generalizability of DDA is built upon two foundations: 1) **VAE artifacts generalize across generators.** Because VAE-reconstructed images are the closest synthetic counterparts to real images, decision boundaries learned from these pairs are likely to remain effective for distinguishing other, more distant synthetic variants (e.g., those from text-to-image generators). Moreover, since the VAE decoder is typically the final stage in diffusion-based generators, its artifacts are less influenced by subsequent modules. 2) **Dual-domain alignment mitigates dataset bias.** By aligning synthetic images with real ones in both the frequency and pixel domains, DDA reduces real-synthetic discrepancies more effectively than alternative reconstruction-based methods. In particular, it eliminates high-frequency bias—commonly introduced by compression or generative artifacts—leading to stronger generalization and reduced reliance on spurious features.

**Comparison to Dataset Alignment Methods.** We validate that DDA creates the closest real-synthetic image pairs when compared to other alignment methods from the following three viewpoints: 1) Pixel domain: Left of Figure 7 shows that DDA-aligned images lead to minimal MSE loss compared to the original image; 2) Frequency domain: Right of Figure 7 shows that DDA-aligned images are most similar to the original image in frequency space; 3) Feature domain: Figure 8 validates that the cluster center of DDA-aligned images is closest to the center of real images.

## 4 Experiments

### 4.1 Experimental Setup

**Datasets** All compared detectors are evaluated on eleven diverse datasets, including seven benchmark datasets (GenImage [59], DRCT-2M [4], Synthbuster [1], DDA-COCO, EvalGEN, AIGCDetectionBenchmark [56] and ForenSynths [43] ) and four in-the-wild datasets (Chameleon [48], WildRF [2], SynthWildx [7] and BFree-Online [14]), where images are sourced from the web. These datasets contain real images from different sources and various generators, including diffusion models, GAN models, auto-regressive models, and other unknown models. They differ in format, content, and resolution, thereby minimizing evaluation bias. Table 1 outlines the datasets' details.

Table 1: Overview of the evaluation benchmarks. "SD" denotes Stable Diffusion and "AR" denotes auto-regressive models. The diversity of data sources and generator types—along with four benchmarks collected from in-the-wild data with unknown generators and post-processing—ensures that the overall evaluation more accurately reflects a detector's generalizability and practical applicability.

| Dataset | Real/Fake | Source | #Models | Model Types |
|---|---|---|---|---|
| DDA-COCO (ours) | 5K/25K | MSCOCO | 5 | SD |
| EvalGEN (ours) | 0/2765 | Prompt | 5 | SD & AR |
| GenImage [59] | 48K/48K | ImageNet | 8 | SD & GAN |
| DRCT-2M [4] | 5K/80K | MSCOCO | 16 | SD |
| Synthbuster [1] | 1K/9K | RAISE | 9 | SD |
| AIGCDetectionBenchmark [56] | 76.25K/76.25K | LSUN & MSCOCO & ImageNet & CelebA & FFHQ | 17 | SD & GAN |
| ForenSynths [43] | 36.2K/36.2K | LSUN & MSCOCO & ImageNet & CelebA & others | 11 | GAN |
| Chameleon [48] | 14.9K/11.2K | Internet | unknown | unknown |
| WildRF [2] | 500/500 | Reddit, FB, X | unknown | unknown |
| SynthWildx [7] | 500/1.5K | X | 3 | SD |
| BFree-Online [14] | 303/641 | Internet | unknown | unknown |

Table 2: **Overall comparison across 11 benchmarks.** To ensure fairness and reproducibility, we use official checkpoints released by each method. We exclude B-Free [14] from this comparison due to the unavailability of public code. JPEG compression with a quality factor of 96 is applied to the synthetic images in GenImage, ForenSynths, and AIGCDetectionBenchmark to mitigate format bias. The number of generators used in each dataset is reported below the dataset name, where G refers to GANs, D to Diffusion models, and AR to Auto-Regressive models. Bold numbers indicate the best performance per column; underlined numbers indicate the second-best.

| Method | Manually Curated Datasets | | | | | | | In-the-Wild Datasets | | | | Avg | Min |
|---|---|---|---|---|---|---|---|---|---|---|---|---|---|
| | GenImage 1G + 7D | DRCT-2M 16D | DDA-COCO 5D | EvalGEN 3D + 2AR | Synthbuster 9D | ForenSynths 11G | AIGCDetection Benchmark 7G + 10D | Chameleon Unknown | Synthwildx 3D | WildRF Unknown | Bfree-Online Unknown | | |
| NPR (CVPR'24) [8] | 51.5 ± 6.3 | 37.3 ± 15.0 | 42.2 ± 5.4 | 2.9 ± 2.7 | 50.0 ± 2.6 | 47.9 ± 22.6 | 53.1 ± 12.2 | 59.9 | 49.8 ± 10.0 | 63.5 ± 13.6 | 49.5 | 46.1 ± 16.1 | 2.9 |
| UnivFD (CVPR'23) [32] | 64.1 ± 10.8 | 61.8 ± 8.9 | 52.4 ± 1.5 | 15.4 ± 14.2 | 67.8 ± 14.4 | 77.7 ± 16.1 | 72.5 ± 17.3 | 50.7 | 52.3 ± 11.3 | 55.3 ± 5.7 | 49.0 | 56.3 ± 16.5 | 15.4 |
| FatFormer (CVPR'24) [31] | 62.8 ± 10.4 | 52.2 ± 5.7 | 51.7 ± 1.5 | 45.6 ± 33.1 | 56.1 ± 10.7 | 90.0 ± 11.8 | 85.0 ± 14.9 | 51.2 | 52.1 ± 8.2 | 58.9 ± 8.0 | 50.0 | 59.6 ± 14.6 | 45.6 |
| SAFE (KDD'25) [23] | 50.3 ± 1.2 | 59.3 ± 19.2 | 49.9 ± 0.3 | 1.1 ± 0.6 | 46.5 ± 20.8 | 49.7 ± 2.7 | 50.3 ± 1.1 | 59.2 | 49.1 ± 0.7 | 57.2 ± 18.5 | 50.5 | 47.6 ± 16.0 | 1.1 |
| C2P-CLIP (AAAI'25) [38] | 74.4 ± 8.4 | 59.2 ± 9.9 | 51.3 ± 0.6 | 38.9 ± 31.2 | 68.5 ± 11.4 | 92.0 ± 10.1 | 81.4 ± 15.6 | 51.1 | 57.1 ± 4.2 | 59.6 ± 7.7 | 50.0 | 62.1 ± 15.6 | 38.9 |
| AIDE (ICLR'25) [48] | 61.2 ± 11.9 | 64.6 ± 11.8 | 50.0 ± 0.4 | 19.1 ± 11.1 | 53.9 ± 18.6 | 59.4 ± 24.6 | 63.6 ± 13.9 | 63.1 | 48.8 ± 0.8 | 58.4 ± 12.9 | 53.1 | 54.1 ± 12.8 | 19.1 |
| DRCT (ICML'24) [4] | 84.7 ± 2.7 | 90.5 ± 7.4 | 60.2 ± 4.3 | 77.8 ± 5.4 | 84.8 ± 3.6 | 73.9 ± 13.4 | 81.4 ± 12.2 | 56.6 | 55.1 ± 1.8 | 50.6 ± 3.5 | 55.7 | 70.1 ± 14.6 | 50.6 |
| AlignedForensics (ICLR'25) [33] | 79.0 ± 22.7 | 95.5 ± 6.1 | 86.5 ± 19.1 | 68.0 ± 20.7 | 77.4 ± 25.0 | 53.9 ± 7.1 | 66.6 ± 21.6 | 71.0 | 78.8 ± 17.8 | 80.1 ± 10.3 | 68.5 | 75.0 ± 11.1 | 53.9 |
| **DDA (ours)** | **91.7 ± 7.8** | **98.1 ± 1.4** | **92.2 ± 10.6** | **97.2 ± 4.2** | **90.1 ± 5.6** | 81.4 ± 13.9 | **87.8 ± 12.6** | **82.4** | **90.9 ± 3.1** | **90.3 ± 3.5** | **95.1** | **90.7 ± 5.3** | **81.4** |

**DDA-COCO and EvalGEN** DDA-COCO consists of five subsets containing reconstructed images of MSCOCO [28] validation set by different VAEs, utilizing frequency-level alignment. We construct the EvalGEN dataset using the five latest text-to-image (T2I) generators using aligned prompts from the GenEval benchmark [11]. Notably, **we are the first work to involve auto-regressive-based T2I generators for image forensics** in the AIGI detection field. Specifically, we introduce each generator as follows: (1) **Flux** [22]: the SOTA diffusion-based generator, offering extremely higher-resolution output images. (2) **GoT** [10]: A multimodal model combining LLM and diffusion processes to enable reasoning-guided image generation. (3) **Infinity** [16]: A bitwise auto-regressive model using infinite-vocabulary tokenization and self-correction for faster and higher-fidelity image generation. (4) **OmiGen** [44]: A unified multimodal framework capable of handling diverse image generation tasks within a single, simplified architecture. (5) **NOVA** [9]: A non-quantized auto-regressive model designed for efficient image and video generation, achieving high fidelity with reduced computational overhead. These models allow our **EvalGEN to serve as a very high-quality benchmark** for evaluating the generalizability of detectors on unseen generators.

**Implementation Details** We use DINOv2 as the backbone and fine-tune it with LoRA, using a rank of 8. The input resolution is set to 336×336, employing random cropping during training and center cropping during validation. Padding is applied when the image height or width is insufficient. The training data exclusively consists of MSCOCO [29] images and their DDA-aligned counterparts. During VAE reconstruction, to ensure that the reconstructed image size matches the real one, we first center-crop each image to the largest size that is a multiple of 8, following the VAE model's design. For frequency alignment in DDA, we apply the same JPEG compression to each reconstructed counterpart with a 50% probability during training, allowing the model to encounter both JPEG and PNG formats of synthetic images. *All evaluations are conducted using a single model without any dataset-specific fine-tuning or threshold adjustments.*

**Evaluation Metrics and Comparative Methods** Unless otherwise specified, we report balanced accuracy, the average of real and fake accuracies, as the evaluation metric, following works [8, 32, 31, 23, 38, 48, 4, 33, 14]. The methods compared include four frequency-based detectors: NPR [8], SAFE [23], and AIDE [48]; three CLIP-based detectors: UnivFD [32], Fatformer [31], and C2P-CLIP [38]; and two data alignment methods: DRCT [4] and AlignedForensics [33].

Table 3: Comparison of balanced accuracy between DDA and compared methods on DRCT-2M.

| Method | LDM | SDv1.4 | SDv1.5 | SDv2 | SDXL | SDXL-Refiner | SD-Turbo | SDXL-Turbo | LCM-SDv1.5 | LCM-SDXL | SDv1-Ctrl | SDv2-Ctrl | SDXL-Ctrl | SDv1-DR | SDv2-DR | SDXL-DR | Avg. |
|---|---|---|---|---|---|---|---|---|---|---|---|---|---|---|---|---|---|
| NPR (CVPR'24) [8] | 33.0 | 29.1 | 29.0 | 35.1 | 33.2 | 28.4 | 27.9 | 27.9 | 29.4 | 30.2 | 28.4 | 28.3 | 34.7 | 67.9 | 67.4 | 66.1 | 37.3 ± 15.0 |
| UnivFD (CVPR'23) [32] | 85.4 | 56.8 | 56.4 | 58.2 | 63.2 | 55.0 | 56.5 | 53.0 | 54.5 | 65.9 | 68.0 | 65.4 | 75.9 | 64.6 | 56.2 | 53.9 | 61.8 ± 8.9 |
| FatFormer (CVPR'24) [31] | 55.9 | 48.2 | 48.2 | 48.2 | 48.2 | 48.3 | 48.2 | 48.2 | 48.3 | 50.6 | 49.7 | 49.9 | 59.8 | 66.3 | 60.6 | 56.0 | 52.2 ± 5.7 |
| SAFE (KDD'25) [23] | 50.3 | 50.1 | 50.0 | 50.0 | 49.9 | 50.1 | 50.0 | 50.0 | 50.1 | 50.0 | 49.9 | 50.0 | 54.7 | 98.2 | 98.5 | 97.3 | 59.3 ± 19.2 |
| C2P-CLIP (AAAI'25) [38] | 83.0 | 51.7 | 51.7 | 52.9 | 51.9 | 64.6 | 51.7 | 50.6 | 52.0 | 66.1 | 56.9 | 54.7 | 77.8 | 67.2 | 57.1 | 56.7 | 59.2 ± 9.9 |
| AIDE (ICLR'25) [48] | 64.4 | 74.9 | 75.1 | 58.5 | 53.5 | 66.3 | 52.8 | 52.8 | 70.0 | 54.3 | 65.9 | 53.6 | 53.9 | 95.3 | 73.3 | 69.0 | 64.6 ± 11.8 |
| DRCT (ICML'24) [4] | 96.7 | 96.3 | 96.3 | 94.9 | 96.2 | 93.5 | 93.4 | 92.9 | 91.2 | 95.0 | 95.6 | 92.7 | 92.0 | 94.1 | 69.6 | 57.4 | 90.5 ± 7.4 |
| AlignedForensics (ICLR'25) [33] | 99.9 | 99.9 | 99.9 | 99.6 | 90.2 | 81.3 | 99.7 | 89.4 | 99.7 | 90.0 | 99.9 | 99.2 | 87.6 | 99.9 | 99.8 | 92.6 | 95.5 ± 6.1 |
| **DDA (ours)** | 99.2 | 98.9 | 99.0 | 98.3 | 98.0 | 96.8 | 97.9 | 94.8 | 95.9 | 98.2 | 98.7 | 99.0 | 99.4 | 99.0 | 99.5 | 96.3 | **98.1 ± 1.4** |

Table 4: Comparison of balanced accuracy on GenImage.

| Method | Midjourney | SDv1.4 | SDv1.5 | ADM | GLIDE | Wukong | VQDM | BigGAN | Avg. |
|---|---|---|---|---|---|---|---|---|---|
| NPR (CVPR'24) [8] | 53.4 | 55.1 | 55.0 | 43.8 | 41.2 | 57.4 | 48.4 | 57.7 | 51.5 ± 6.3 |
| UnivFD (CVPR'23) [32] | 55.1 | 55.6 | 55.7 | 62.5 | 61.3 | 61.1 | 76.9 | 84.4 | 64.1 ± 10.8 |
| FatFormer (CVPR'24) [31] | 52.1 | 53.6 | 53.8 | 61.4 | 65.5 | 60.9 | 72.5 | 82.2 | 62.8 ± 10.4 |
| SAFE (KDD'25) [23] | 49.0 | 49.7 | 49.8 | 49.5 | 53.0 | 50.3 | 50.2 | 50.9 | 50.3 ± 1.2 |
| C2P-CLIP (AAAI'25) [38] | 56.6 | 77.5 | 76.9 | 71.6 | 73.5 | 79.4 | 73.7 | 85.9 | 74.4 ± 8.4 |
| AIDE (ICLR'25) [48] | 58.2 | 77.2 | 77.4 | 50.4 | 54.6 | 70.5 | 50.8 | 50.6 | 61.2 ± 11.9 |
| DRCT (ICML'24) [4] | 82.4 | 88.3 | 88.2 | 76.9 | 86.1 | 87.9 | 85.4 | 87.0 | 84.7 ± 2.7 |
| AlignedForensics (ICLR'25) [33] | 97.5 | 99.7 | 99.6 | 52.4 | 57.6 | 99.6 | 75.0 | 50.6 | 79.0 ± 22.7 |
| **DDA (ours)** | 95.6 | 98.7 | 98.6 | 89.5 | 89.6 | 98.7 | 76.5 | 86.5 | **91.7 ± 7.8** |

Table 5: Comparison of balanced accuracy on AIGCDetectionBenchmark.

| Method | ADM | DALLE2 | GLIDE | Midjourney | VQDM | BigGAN | CycleGAN | GauGAN | ProGAN | SDXL | SD14 | SD15 | StarGAN | StyleGAN | StyleGAN2 | WFR | Wukong | Avg. |
|---|---|---|---|---|---|---|---|---|---|---|---|---|---|---|---|---|---|---|
| NPR (CVPR'24) [8] | 43.8 | 20.0 | 41.2 | 53.4 | 48.4 | 53.1 | 76.6 | 42.2 | 58.7 | 59.6 | 55.1 | 55.0 | 67.4 | 57.9 | 54.6 | 58.8 | 57.4 | 53.1 ± 12.2 |
| UnivFD (CVPR'23) [32] | 62.5 | 50.0 | 61.3 | 55.1 | 76.9 | 87.5 | 96.9 | 98.8 | 99.4 | 58.2 | 55.6 | 55.7 | 95.1 | 80.0 | 69.4 | 69.2 | 61.1 | 72.5 ± 17.3 |
| FatFormer (CVPR'24) [31] | 80.2 | 68.5 | 91.1 | 54.4 | 88.0 | 99.2 | 99.5 | 99.1 | 98.5 | 71.7 | 67.5 | 67.2 | 99.4 | 98.0 | 98.8 | 75.6 |  | 85.0 ± 14.9 |
| SAFE (KDD'25) [23] | 49.5 | 49.5 | 53.0 | 49.0 | 50.2 | 52.2 | 51.9 | 50.0 | 50.0 | 49.8 | 49.7 | 49.8 | 50.1 | 50.0 | 50.0 | 49.8 | 50.3 | 50.3 ± 1.1 |
| C2P-CLIP (AAAI'25) [38] | 71.6 | 52.3 | 73.5 | 56.6 | 73.7 | 98.4 | 96.8 | 98.8 | 99.3 | 62.3 | 77.5 | 76.9 | 99.6 | 93.1 | 79.4 | 94.8 | 79.4 | 81.4 ± 15.6 |
| AIDE (ICLR'25) [48] | 52.9 | 51.1 | 60.2 | 49.8 | 69.3 | 70.1 | 93.6 | 60.6 | 89.0 | 49.6 | 51.6 | 51.0 | 72.1 | 66.5 | 59.0 | 80.6 | 54.5 | 63.6 ± 13.9 |
| DRCT (ICML'24) [4] | 79.9 | 89.2 | 89.2 | 85.5 | 88.6 | 81.4 | 91.0 | 93.8 | 71.1 | 88.3 | 91.4 | 91.0 | 53.0 | 62.7 | 63.8 | 73.9 | 90.8 | 81.4 ± 12.2 |
| AlignedForensics (ICLR'25) [33] | 51.6 | 52.0 | 55.6 | 96.2 | 72.1 | 51.2 | 49.5 | 50.8 | 50.7 | 95.1 | 99.7 | 99.6 | 53.8 | 52.7 | 51.6 | 50.0 | 99.6 | 66.6 ± 21.6 |
| **DDA (ours)** | 89.5 | 94.6 | 89.6 | 95.6 | 76.6 | 91.0 | 72.5 | 92.7 | 92.8 | 99.4 | 98.7 | 98.6 | 72.7 | 87.8 | 90.2 | 52.1 | 98.8 | **87.8 ± 12.6** |

Table 6: Comparison of balanced accuracy between DDA and compared methods on ForenSynths.

| Method | BigGAN | CRN | CycleGAN | DeepFake | GauGAN | IMLE | ProGAN | SAN | SeeingDark | StarGAN | StyleGAN | StyleGAN 2 | WFR | Avg. |
|---|---|---|---|---|---|---|---|---|---|---|---|---|---|---|
| NPR (CVPR'24) [8] | 53.1 | 0.4 | 76.6 | 35.7 | 42.2 | 5.3 | 58.7 | 48.4 | 63.6 | 67.4 | 57.9 | 54.6 | 58.8 | 47.9 ± 22.6 |
| UnivFD (CVPR'23) [32] | 87.5 | 55.7 | 96.9 | 69.4 | 98.8 | 68.1 | 99.4 | 58.2 | 62.2 | 95.1 | 80.0 | 69.4 | 69.2 | 77.7 ± 16.1 |
| FatFormer (CVPR'24) [31] | 99.3 | 72.1 | 99.5 | 93.0 | 99.3 | 72.1 | 98.4 | 70.8 | 81.9 | 99.4 | 98.1 | 98.9 | 88.3 | 90.1 ± 11.8 |
| SAFE (KDD'25) [23] | 52.2 | 50.0 | 51.9 | 50.1 | 50.0 | 50.0 | 50.0 | 50.9 | 41.1 | 50.1 | 50.0 | 50.0 | 49.8 | 49.7 ± 2.7 |
| C2P-CLIP (AAAI'25) [38] | 98.4 | 93.3 | 96.8 | 92.6 | 93.2 | 99.3 | 63.2 | 94.7 | 99.6 | 93.1 | 79.4 | 94.8 |  | 92.1 ± 10.1 |
| AIDE (ICLR'25) [48] | 70.1 | 12.2 | 93.6 | 53.2 | 60.6 | 15.9 | 89.0 | 55.3 | 44.2 | 72.1 | 66.5 | 59.0 | 80.6 | 59.4 ± 24.6 |
| DRCT (ICML'24) [4] | 81.4 | 78.4 | 91.0 | 51.5 | 93.8 | 82.6 | 71.1 | 84.9 | 72.2 | 53.0 | 62.7 | 63.8 | 73.9 | 73.9 ± 13.4 |
| AlignedForensics (ICLR'25) [33] | 51.2 | 50.4 | 49.5 | 71.7 | 50.8 | 49.7 | 50.7 | 67.6 | 51.4 | 53.8 | 52.7 | 51.6 | 50.0 | 53.9 ± 7.1 |
| **DDA (ours)** | 91.0 | 87.0 | 72.5 | 76.5 | 92.7 | 89.7 | 92.8 | 94.7 | 58.6 | 72.7 | 87.8 | 90.2 | 52.1 | 81.4 ± 13.9 |

## 4.2 Cross-Dataset and Cross-Model Comparison

**Overall Comparison on 11 Benchmarks.** Table 2 presents a comprehensive comparison across 11 datasets—7 manually curated and 4 in-the-wild—covering most known open-source AIGI evaluation benchmarks. To the best of our knowledge, the first large-scale comparison of its kind. As benchmarks vary greatly, the benefits of exploiting any bias are minimized, making the average accuracy across the 11 benchmarks more representative of the detector's practical performance. The results show that: 1) DDA achieves an average accuracy of 90.7%, marking a 15.7% improvement over the second-best method. Notably, DDA reaches 82.4% accuracy on the challenging Chameleon benchmark, where only AlignedForensics achieves above 70%; 2) DDA also has the highest minimal accuracy of 81.4%, significantly outperforming the second-best method by 27.5%. Moreover, DDA exhibits the smallest deviation across benchmarks, less than half of the other methods, suggesting it is more reliable and robust; 3) An interesting observation is that, when comparing methods across datasets, detectors trained on more aligned data tend to achieve much higher average accuracy. The degree of data alignment in the detectors, in increasing order, is as follows: UnivFD (no data alignment), DRCT (data alignment via diffusion reconstruction), AlignedForensics (data alignment via VAE reconstruction), and DDA (dual-domain data alignment). The overall average accuracy follows the same order as the degree of data alignment. This clearly demonstrates the effectiveness of data alignment in improving a detector's generalizability, supporting our previous assertion that data alignment helps models learn more transferable decision boundariess.

**Detailed Comparison on DRCT-2M, GenImage, AIGCDetectionBenchmark, ForenSynths, Synthbuster, SynthWildx, and WildRF.** Table 3 to Table 8 report the detailed performance of various methods across these subsets. From the results, we observe the following: (1) Consistent superiority: DDA not only surpasses other detectors by a substantial margin in average accuracy (ranging from 3% to 10%) but also achieves consistently lower deviations across all benchmarks. Given the diversity of real image sources and the inclusion of both GAN- and diffusion-based models,

Table 7: Comparison of balanced accuracy between DDA and compared methods on Synthbuster.

| Method | DALL·E 2 | DALL·E 3 | Firefly | GLIDE | Midjourney | SD 1.3 | SD 1.4 | SD 2 | SDXL | Avg. |
|---|---|---|---|---|---|---|---|---|---|---|
| NPR (CVPR'24) [8] | 51.1 | 49.3 | 46.5 | 48.5 | 52.8 | 51.4 | 51.8 | 46.0 | 52.8 | 50.0 ± 2.6 |
| UnivFD (CVPR'23) [32] | 83.5 | 47.4 | 89.9 | 53.3 | 52.5 | 70.4 | 69.9 | 75.7 | 68.0 | 67.8 ± 14.4 |
| FatFormer (CVPR'24) [31] | 59.4 | 39.5 | 60.3 | 72.7 | 44.4 | 53.7 | 54.0 | 52.3 | 69.1 | 56.1 ± 10.7 |
| SAFE (KDD'25) [23] | 58.0 | 9.9 | 10.3 | 52.2 | 56.7 | 59.4 | 59.1 | 53.0 | 59.5 | 46.5 ± 20.8 |
| C2P-CLIP (AAAI'25) [38] | 55.6 | 63.2 | 59.5 | 86.7 | 52.9 | 75.2 | 76.7 | 69.2 | 77.7 | 68.5 ± 11.4 |
| AIDE (ICLR'25) [48] | 34.9 | 33.7 | 24.8 | 65.0 | 57.5 | 74.1 | 73.7 | 53.2 | 68.4 | 53.9 ± 18.6 |
| DRCT (ICML'24) [4] | 77.2 | 86.6 | 84.1 | 82.6 | 73.7 | 86.6 | 86.6 | 83.2 | 71.3 | 84.8 ± 3.6 |
| AlignedForensics (ICLR'25) [33] | 50.2 | 48.9 | 51.7 | 53.5 | 98.7 | 98.8 | 98.8 | 98.6 | 97.3 | 77.4 ± 25.0 |
| **DDA (ours)** | **86.3** | **90.0** | **91.9** | 76.5 | 93.5 | 92.9 | 92.7 | 93.3 | 93.5 | **90.1 ± 5.6** |

Table 8: Comparison of balanced accuracy between DDA and compared methods on SynthWildx and WildRF.

| Method | SynthWildx | | | | WildRF | | | |
|---|---|---|---|---|---|---|---|---|
| | DALL·E 3 | Firefly | Midjourney | Avg. | Facebook | Reddit | Twitter | Avg. |
| NPR (CVPR'24) [8] | 43.6 | 61.3 | 44.5 | 49.8 ± 10.0 | 78.1 | 61.0 | 51.3 | 63.5 ± 13.6 |
| UnivFD (CVPR'23) [32] | 45.4 | 65.3 | 46.2 | 52.3 ± 11.3 | 49.1 | 60.2 | 56.5 | 55.3 ± 5.7 |
| FatFormer (CVPR'24) [31] | 46.5 | 61.6 | 48.3 | 52.1 ± 8.2 | 54.1 | 68.1 | 54.4 | 58.9 ± 8.0 |
| SAFE (KDD'25) [23] | 49.4 | 48.2 | 49.6 | 49.1 ± 0.7 | 50.9 | 74.1 | 37.5 | 57.2 ± 18.5 |
| C2P-CLIP (AAAI'25) [38] | 56.9 | 61.4 | 53.0 | 57.1 ± 4.2 | 54.4 | 68.4 | 55.9 | 59.6 ± 7.7 |
| AIDE (ICLR'25) [48] | 63.4 | 48.8 | 51.9 | 48.8 ± 0.8 | 57.8 | 71.5 | 45.8 | 58.4 ± 12.9 |
| DRCT (ICML'24) [4] | 58.3 | 56.4 | 50.5 | 55.1 ± 1.8 | 46.6 | 53.1 | 55.2 | 50.6 ± 3.5 |
| AlignedForensics (ICLR'25) [33] | 85.5 | 58.5 | 92.2 | 78.8 ± 17.8 | 89.4 | 69.1 | 81.8 | 80.1 ± 10.3 |
| **DDA (ours)** | **92.3** | **87.3** | **93.1** | **90.9 ± 3.1** | **93.1** | **86.4** | **91.5** | **90.3 ± 3.5** |

Table 9: Comparison of balanced accuracy between our DDA and other methods on DDA-COCO.

| Method | real | fake | | | | | | Avg |
|---|---|---|---|---|---|---|---|---|
| | | XL | EMA | MSE | SD21 | SD35 | FLUX.1 | |
| NPR (CVPR'24) [8] | 55.4 | 16.1 | 31.1 | 41.3 | 41.2 | 24.9 | 19.2 | 42.2 ± 5.4 |
| UnivFD (CVPR'23) [32] | 99.2 | 3.9 | 9.6 | 7.3 | 7.4 | 3.4 | 1.8 | 52.4 ± 1.5 |
| FatFormer (CVPR'24) [31] | 96.4 | 5.4 | 6.9 | 10.4 | 10.3 | 6.6 | 2.8 | 51.7 ± 1.5 |
| SAFE (KDD'25) [23] | 98.8 | 0.6 | 0.9 | 0.9 | 1.0 | 0.3 | 1.8 | 49.9 ± 0.3 |
| C2P-CLIP (AAAI'25) [38] | 99.5 | 2.0 | 2.7 | 4.3 | 4.2 | 4.0 | 1.3 | 51.3 ± 0.6 |
| AIDE (ICLR'25) [48] | 98.8 | 0.6 | 2.2 | 1.7 | 1.8 | 0.3 | 0.6 | 50.0 ± 0.4 |
| DRCT (ICML'24) [4] | 94.2 | 16.9 | 34.8 | 33.5 | 33.6 | 21.7 | 17.2 | 60.2 ± 4.3 |
| AlignedForensics (ICLR'25) [33] | 99.8 | 82.5 | 99.2 | 99.0 | 99.1 | 55.4 | 3.6 | 86.5 ± 19.1 |
| **DDA (ours)** | 99.0 | 95.0 | 99.3 | 99.7 | 99.7 | 68.1 | 50.2 | **92.2 ± 10.6** |

Table 10: Comparison of balanced accuracy on EvalGEN.

| Method | Flux | GoT | Infinity | NOVA | OmiGen | Avg. |
|---|---|---|---|---|---|---|
| NPR (CVPR'24) [8] | 0.7 | 0.2 | 6.5 | 4.7 | 2.2 | 2.9 ± 2.7 |
| UnivFD (CVPR'23) [32] | 4.0 | 9.2 | 15.7 | 8.3 | 39.6 | 15.4 ± 14.2 |
| FatFormer (CVPR'24) [31] | 9.9 | 47.9 | 44.7 | 98.3 | 27.3 | 45.6 ± 33.1 |
| SAFE (KDD'25) [23] | 1.0 | 0.5 | 1.9 | 0.6 | 1.6 | 1.1 ± 0.6 |
| C2P-CLIP (AAAI'25) [38] | 8.7 | 49.6 | 35.3 | 86.4 | 14.5 | 38.9 ± 31.2 |
| AIDE (ICLR'25) [48] | 17.9 | 24.7 | 3.4 | 16.3 | 33.4 | 19.1 ± 11.1 |
| DRCT (ICML'24) [4] | 72.5 | 81.4 | 77.9 | 84.6 | 72.5 | 77.8 ± 5.4 |
| AlignedForensics (ICLR'25) [33] | 32.0 | 72.3 | 74.0 | 84.8 | 77.0 | 68.0 ± 20.7 |
| **DDA (ours)** | **89.9** | **99.5** | **97.8** | **99.5** | **99.5** | **97.2 ± 4.2** |

these results strongly demonstrate the effectiveness and generalizability of DDA. (2) Exception on ForenSynths: DDA underperforms slightly on the ForenSynths benchmark. We attribute this to the fact that the two methods outperforming DDA—FatFormer and C2P-CLIP—were trained on ProGAN, which is also the generator used in the ForenSynths subsets, giving them an advantage. Moreover, some data in the ForenSynths are generated by older and smaller models, which deviate significantly from modern generators, contributing to DDA's performance degradation.

**Comparison on DDA-COCO and EvalGEN**  Table 9 and Table 10 report accuracies on our two proposed benchmarks, DDA-COCO and EvalGEN, respectively. We observe the following: (1) Generalization across diffusion models: DDA, trained solely on SD 2.1–reconstructed data, generalizes well to other diffusion models and exhibits a smaller standard deviation. This suggests that DDA learns a universal upsampling artifact shared across diverse generative models, reinforcing our claim that data alignment enhances generalizability. (2) Effectiveness of data alignment: Results on DDA-COCO highlight the importance of data alignment. Methods lacking explicit alignment—such as NPR [8], UnivFD [32], FatFormer [31], C2P-CLIP [38], AIDE [48], and SAFE [23]—exhibit large disparities between real and fake accuracies, revealing underlying dataset biases. (3) Performance on emerging generators: DDA also achieves SOTA on EvalGEN, excelling on auto-regressive generators, further validating its strong cross-architecture generalizability.

**Comparison on Generation Time Cost**  We compare three methods which introduce train data generation—DRCT [4], AlignedForensics [33], and B-Free [14]. Table 11 presents the number of real and synthetic images used for training, generation method and the estimated reconstruction time (Single Image & Full Set) using each method, which is tested by generating 100 synthetic images. Results show that our DDA requires the least amount of training data and reconstruction time, confirming its effectiveness and efficiency in terms of training cost.

Table 11: Comparing on data generation time.

| Method | # Real / Fake | Generation Method | Time Per Image | Full Construction Time |
|---|---|---|---|---|
| DRCT | 118K / 354K | Diff. Rec. | 0.6569 ± 0.0050 s | 64.6 h |
| AlignedForensics | 179K / 179K | VAE Rec. | 0.1756 ± 0.0692 s | 8.73 h |
| B-Free | 51K / 309K | Diff. Rec + Inpaint. | 3.0150 ± 0.0125 s | 258.79 h |
| **DDA (ours)** | 118K / 118K | VAE Rec. + DDA | 0.1792 ± 0.0704 s | **5.9 h** |

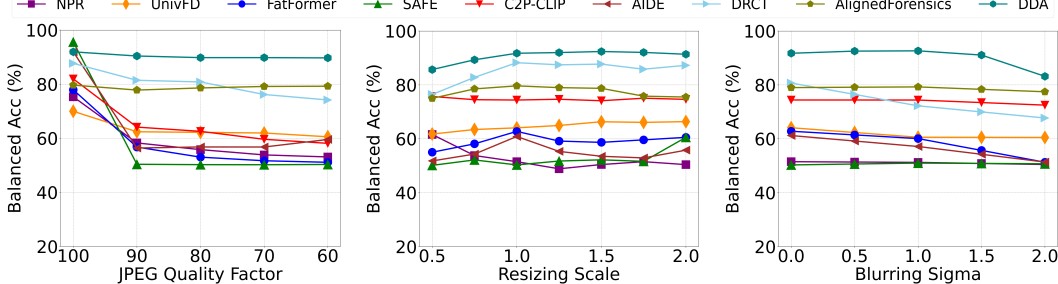

Figure 9: Robustness analysis on GenImage.

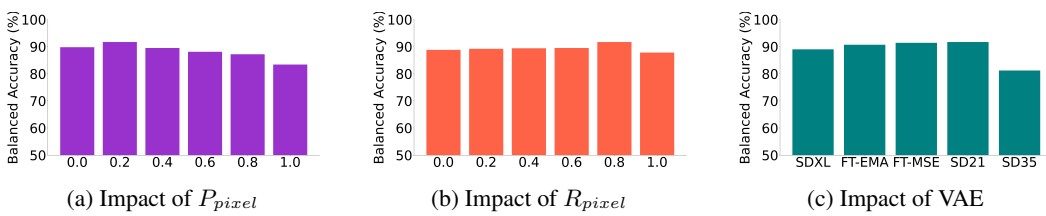

(a) Impact of $P_{pixel}$       (b) Impact of $R_{pixel}$       (c) Impact of VAE

Figure 10: **Ablation studies.** (a) $P_{pixel}$: probability of applying pixel-level alignment; (b) $R_{pixel}$: upper bound for sampling the pixel mixup ratio; (c) VAE: backbone used for training data reconstruction. Results show the impact of each hyperparameter on the performance of DDA.

### 4.3 Evaluation on Robustness

Figure 9 shows the results of three robustness evaluations on the GenImage-JPEG96 dataset for all compared methods. Results show that: (1) DDA shows strong robustness across all three post-processing methods, outperforming the second-best method by 10.5%, 4.1%, and 5.7% under JPEG 60, RESIZE 2.0, and BLUR 2.0, respectively. (2) Methods lacking alignment, such as NPR [8], SAFE [23], and AIDE [48], demonstrate poor robustness under JPEG compression and resizing. In contrast, methods with alignment perform much better, emphasizing the importance of data alignment.

### 4.4 Ablation Studies

Figure 10 illustrates the impact of $P_{pixel}$, $R_{pixel}$, and the choice of VAE in training data generation. Results indicate that the detector maintains consistent accuracy when $P_{pixel}$ and $R_{pixel}$ are between 0.2 and 0.8, with performance drops observed at 0.0 and 1.0. Experiments with different VAEs confirm that SD21 is the most effective choice.

## 5 Conclusion

In this paper, we demonstrate that pixel-domain alignment alone is insufficient for fully aligning real and synthetic image pairs. Building on this, we propose DDA to align synthetic images with real ones across both pixel and frequency domains, thereby mitigating bias. We also introduce two AIGI benchmarks: DDA-COCO and EvalGEN. Extensive experiments across eleven benchmarks demonstrate the consistent superiority of DDA. We believe that DDA, DDA-COCO, and EvalGEN provide a solid foundation for advancing the generalization of AIGI detection.

**Limitations and Future Work**    While DDA demonstrates strong performance across extensive benchmarks, there remains a gap in its application to real-world scenarios, particularly due to the heavy post-processing applied to images in such contexts. In practice, we observe that even authentic photos taken by smartphones may exhibit synthetic-like artifacts, likely resulting from the AI-based enhancements embedded in modern smartphone camera pipelines, which adds complexity to real-world AIGI detection. In future work, we plan to develop a more practical AIGI detector tailored to address these real-world challenges.

# 6 Acknowledgements

The authors would like to express our sincere gratitude to NeurIPS anonymous reviewers for their constructive feedback. This work is supported by the National Natural Science Foundation of China under Grant 62572188 and Grant 62272164.

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

The Appendix provides additional technical and evaluative details of our work. Section A presents the **implementation details** of DDA. Section B summarizes the **peer methods**. Section C **visualizes the degree of data alignment** of different methods in the feature space. Section D presents **ablation studies on input size and backbone**. Section E provides **details of our proposed dataset EvalGEN**, including example prompts used for generation and visualizations of selected samples. Finally, Section F **visualizes the regional detection results of DDA**.

## A  Implementation Details

**Training Details.**   All experiments were conducted on eight NVIDIA V100 GPUs. We trained the detector on a dataset consisting of MSCOCO images and their synthetic counterparts generated through DDA alignment using the VAE from Stable Diffusion 2.1. The model was optimized with a base batch size of 16 and a learning rate of 1e-4. To achieve an effective batch size of 64 without exceeding GPU memory limits, gradient accumulation was applied over four iterations. Balanced accuracy was evaluated on all datasets every 10,000 iterations, and early stopping was employed to prevent overfitting. To help the model better shape its decision boundary, each batch was manually constructed to include both real images and their DDA-aligned counterparts, allowing the model to simultaneously observe closely aligned positive and negative samples.

## B  Peer Methods

Below we provide a brief description of the compared methods used in Section 4 of main paper.

**NPR [8]**   This detector leverages low-level features—neighboring pixel relationships—to distinguish synthetic images from real ones. NPR trains a ResNet-50 to identify upsampling patterns.

**UnivFD [32]**   Instead of conventional supervised training, this method utilizes features from a vision-language model (CLIP-ViT) combined with a linear classifier. This approach avoids overfitting to specific generative artifacts and generalizes better to unseen generators.

**FatFormer [31]**   FatFormer builds on a ViT backbone, incorporating a forgery-aware adapter that adapts features in both the image and frequency domains. It introduces language-guided alignment using contrastive learning with text prompts to improve generalization.

**SAFE [23]**   This method focuses on frequency domain artifacts. The detector is built upon a ResNet backbone and trained with several data augmentation techniques, including random masking.

**C2P-CLIP [38]**   The method utilizes CLIP embeddings with category-specific prompts to enhance deepfake detection generalizability. Image captions are generated using ClipCap and enhanced with category common prompts. During training, these enhanced caption-image pairs train the image encoder through contrastive learning. For inference, only the modified image encoder and a linear classifier are used.

**AIDE [48]**   This work employs a hybrid approach that combines low-level patch statistics with high-level semantics. It uses DCT scoring to select extreme frequency patches for extracting noise patterns through SRM filters, while utilizing CLIP embeddings to capture semantic information. These complementary features are fused through channel-wise concatenation before classification.

**DRCT [4]**   This method reconstructs real images using diffusion models to generate challenging synthetic samples that retain visual content while introducing subtle artifacts. Contrastive learning is employed to guide detectors toward recognizing these fingerprints, improving generalization.

**AlignedForensics [33]**   This method creates aligned datasets by reconstructing real images through a single forward pass in an LDM's autoencoder. This forces the detector to focus exclusively on artifacts introduced by the VAE decoder, avoiding reliance on spurious correlations.

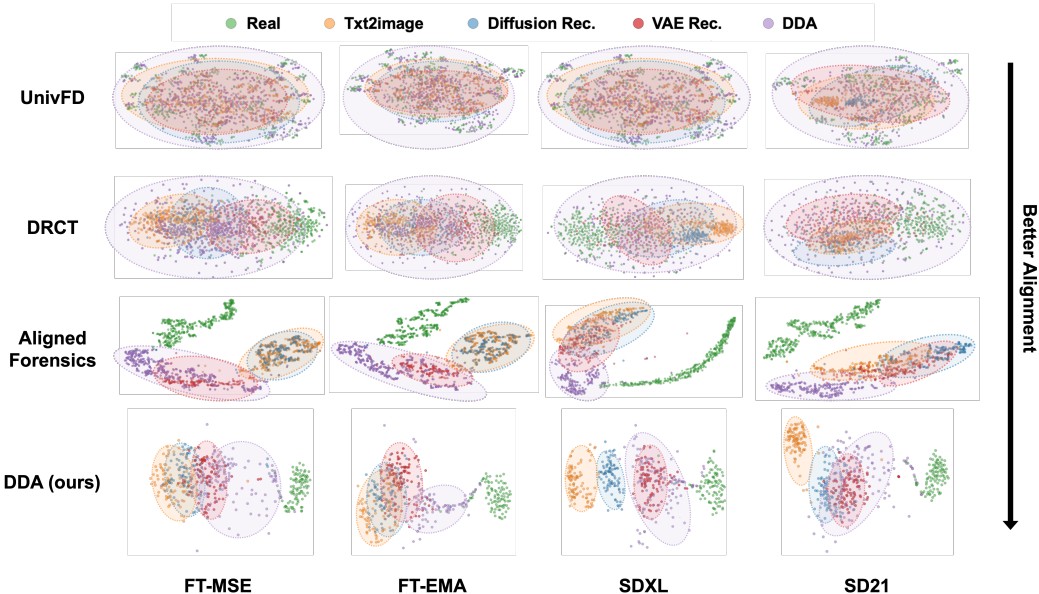

Figure 11: t-SNE visualizations comparing real and synthetic images using detectors trained with different data alignment methods. Rows correspond to the data alignment method used during detector training, while columns represent the generative pipeline—either VAE or diffusion—used to produce synthetic images. **The results show that detectors trained with better dataset alignment are able to separate reconstructed images more distinctly, highlighting the importance of effective dataset alignment in achieving clearer feature space separation.**

**B-Free [14]**   B-Free introduces a training paradigm using self-conditioned diffusion-based reconstructions. It ensures semantic alignment between real and synthetic images so that differences arise solely from generation artifacts. The approach includes content augmentation via inpainting and fine-tunes a DINOv2+reg ViT using large crops to retain forensic signals.

## C    More Comparison Results

**Comparison to Dataset Alignment Methods in Feature Domain**   Fig 11 presents t-SNE visualizations of real and synthetic image features, generated by detectors trained with varying data alignment strategies. Each row, from top to bottom, represents detectors trained on datasets with progressively stronger alignment. **Detectors trained on better-aligned datasets yield more separable feature distributions, suggesting that enhanced alignment facilitates clearer decision boundaries between real and synthetic content.** These findings reinforce the role of data alignment in improving feature separability and overall AGI detector performance.

## D    More Ablation Results

**Ablation on Input Size**   Table 12 presents an ablation study of our method across different input sizes, ranging from 224 to 504. Detectors achieve comparable accuracies across these input sizes.

**Ablation on Backbone**   Table 13 presents an ablation study comparing the performance of different backbone architectures. The ResNet backbone is excluded from this study due to training instability and failure to converge. The relatively poor performance of linear probing methods is attributed to the limited representational capacity of a single linear layer. This observation aligns with the convergence issues observed with ResNet, suggesting that the universal artifacts in our training data are inherently more difficult to learn. In contrast, AlignedForensics [33] successfully employs a ResNet backbone, implying that the artifacts used in our training setup may be subtler or more complex than those captured in prior work. Another key finding is that DINO-LoRA outperforms CLIP-LoRA. This

Table 12: Ablation study across different input sizes.

| Input Size | GenImage | DRCT-2M | EvalGEN | Chameleon | SynthWildx | Avg |
|---|---|---|---|---|---|---|
| 224 | 94.9 | 96.7 | **97.2** | 71.9 | 80.3 | 88.2 ± 11.5 |
| 252 | 95.3 | 96.7 | 94.1 | 72.0 | 84.0 | 88.4 ± 10.5 |
| 280 | **95.7** | 96.2 | 95.4 | 70.1 | 84.6 | 88.4 ± 11.3 |
| 392 | 92.9 | 96.5 | 95.7 | 71.8 | 89.6 | 89.3 ± 10.2 |
| 448 | 93.4 | 97.2 | 89.5 | 65.7 | 89.9 | 87.1 ± 12.4 |
| 504 | 93.0 | 93.0 | 95.8 | 73.2 | 86.2 | 88.2 ± 9.1 |
| **336** | 91.7 | **98.1** | 96.3 | 82.4 | **90.9** | **91.9** ± 6.1 |

Table 13: Ablation on backbones and training strategies. **Linear Probing** refers to training a linear classifier on frozen backbone features. **LoRA Finetune** denotes fine-tuning the backbone using LoRA (rank=8).

| Train Strategy | Backbone | GenImage | DRCT-2M | EvalGEN | Chameleon | SynthWildx | Avg |
|---|---|---|---|---|---|---|---|
| | CLIP ViT-B/16 | 86.4 | 84.2 | 92.3 | 54.7 | 53.9 | 74.3 ± 18.5 |
| | CLIP ViT-B/32 | 83.8 | 80.8 | 97.3 | 63.2 | 54.5 | 75.9 ± 17.1 |
| Linear | CLIP ViT-L/14 | 91.2 | 91.2 | 98.9 | 59.1 | 52.8 | 78.6 ± 21.1 |
| Probing | DINOv2 VIT-S/14 | 68.8 | 74.4 | 59.8 | 60.9 | 62.6 | 65.3 ± 6.2 |
| | DINOv2 VIT-B/14 | 68.3 | 74.5 | 66.2 | 64.1 | 58.6 | 66.8 ± 4.9 |
| | DINOv2 VIT-L/14 | 70.5 | 75.6 | 56.8 | 60.6 | 61.6 | 65.0 ± 7.8 |
| | CLIP ViT-B/16 | 95.2 | 80.3 | 96.2 | 46.6 | 62.0 | 76.1 ± 21.5 |
| LoRA | CLIP ViT-B/32 | 93.2 | 80.6 | 98.5 | 55.0 | 59.0 | 77.3 ± 19.7 |
| Finetune | CLIP ViT-L/14 | **97.0** | 80.4 | **99.2** | 67.7 | 71.8 | 83.2 ± 14.4 |
| | **DINOv2 VIT-L/14** | 91.7 | **98.1** | 96.3 | **82.4** | **90.9** | **91.9** ± 6.1 |

performance difference is likely due to the architectural focus of each backbone: CLIP emphasizes high-level semantic features, while DINO is more attuned to low-level visual patterns—which are more indicative of AI-generated image artifacts. Moreover, DINO-LoRA achieves a lower standard deviation, indicating greater stability for robust AGI detection.

```
Prompt 00 a photo of a backpack
Prompt 01 a photo of a backpack below a cake
Prompt 02 a photo of a backpack right of a sandwich
Prompt 03 a photo of a banana
Prompt 04 a photo of a baseball bat
Prompt 05 a photo of a baseball bat and a bear
Prompt 06 a photo of a baseball bat and a fork
Prompt 07 a photo of a baseball bat and a giraffe
Prompt 08 a photo of a baseball glove
Prompt 09 a photo of a baseball glove and a carrot
Prompt 10 a photo of a baseball glove below an umbrella
Prompt 11 a photo of a baseball glove right of a bear
Prompt 12 a photo of a bear
Prompt 13 a photo of a bear above a clock
Prompt 14 a photo of a bear above a spoon
Prompt 15 a photo of a bed
Prompt 16 a photo of a bed right of a frisbee
Prompt 17 a photo of a bed right of a sports ball
Prompt 18 a photo of a bench
Prompt 19 a photo of a bench and a snowboard
...
```

# E   More Details of EvalGEN

To construct EvalGEN, we used 553 distinct prompts, each generating 20 synthetic images per generator, resulting in 11,060 images per generator and a total of 55,300 synthetic images in the complete dataset. All images are stored in JPEG format with a quality factor of 96. A subset of prompts is provided above to illustrate the dataset's diversity and semantic coverage, while

Fig. 12 shows visual examples from EvalGEN. To balance efficiency and representativeness, for the comparison in Table 2 and Table 10 of the main paper, we selected the first (index 0) image generated for each prompt, yielding 55,300 / 20 = 2,765 samples for quantitative evaluation.

## F    Regional Detection Analysis

Figure 13 displays heatmaps of detection scores across segmented image regions, with numerical overlays indicating the detector's predictions. These results reveal that detection scores vary by region, indicating that synthetic artifacts are spatially uneven. This observation suggests that localized detection strategies could further enhance robustness.

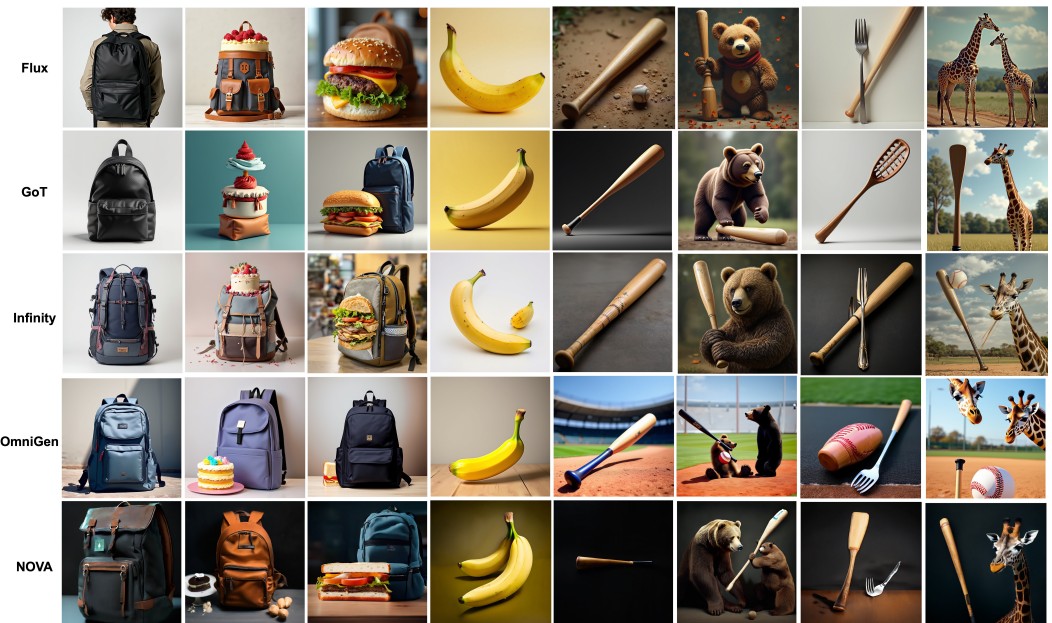

Figure 12: Examples from EvalGEN.

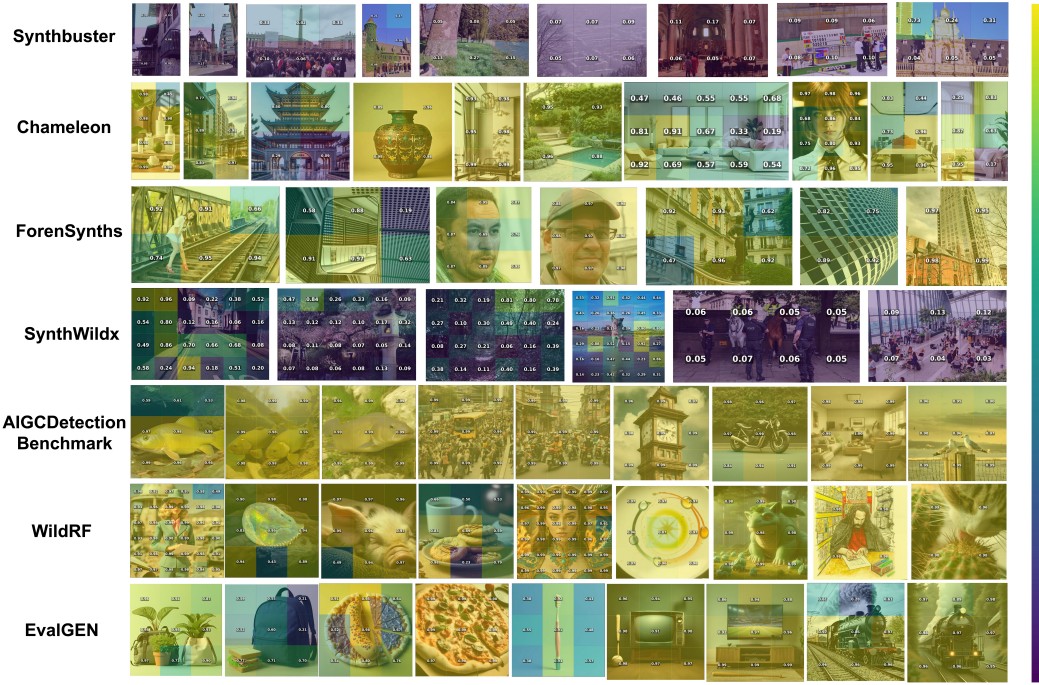

Figure 13: Patch-level detection results. From top to bottom, images are sourced from Synthbuster [1], Chameleon [48], ForenSynths [43], SynthWildx [7], AIGCDetectionBenchmark [56], WildRF [2], and EvalGEN (ours), respectively.

