# OpenReview forum: "Dual Data Alignment Makes AI-Generated Image Detector Easier Generalizable"
_NeurIPS.cc/2025/Conference — NeurIPS 2025 spotlight_

### Official Review · Reviewer_Two9 · 2025-06-21

**Clarity:** 3
**Significance:** 3
**Originality:** 3
**Rating:** 5
**Confidence:** 4

**Summary:**

This paper proposes Dual Data Alignment (DDA), a method to improve the generalization of AI-generated image detectors by aligning real and synthetic images in both pixel and frequency domains. Unlike prior work that only aligns pixel-level content, DDA also corrects frequency-level biases that detectors may exploit. The authors demonstrate that detectors trained on DDA-aligned data significantly outperform baselines across diverse benchmarks, including unseen generative models and real-world datasets. Two new datasets, DDA-COCO and EvalGEN, are also introduced for robust evaluation.

**Questions:**

See the weakness part.

**Ethical Concerns:**

["NO or VERY MINOR ethics concerns only"]

**Final Justification:**

I have read the author's response, and most of the concerns have been addressed. The only remaining concern is the issue raised in Q1 regarding High-Frequency Region Selection. Although the `rectangular-0.5` setting shows performance similar to the `zigzag` method, this choice is not theoretically convincing, as it does not align with the high-frequency distribution of the DCT and may miss some compressed high-frequency components.

I hope the authors will provide a more in-depth discussion or revise the corresponding method in the revised version. Compared to the overall contribution of this paper, this minor concern does not overshadow its merits. I am willing to update my rating to `accept` after discussion. Good luck!

**Limitations:**

See the weakness part.

**Quality:**

3

**Strengths And Weaknesses:**

**Strengths:**

1. Proposes a dual-domain alignment (pixel + frequency), addressing overlooked frequency-level biases in existing methods.
1. Achieves strong generalization across 8 diverse benchmarks, outperforming prior state-of-the-art detectors.
1. The DDA method is efficient to implement and significantly reduces training data generation time.



**Weaknesses:**

1. Since using VAE reconstruction to build training datasets has already been proposed in [1], the main novelty of this paper, to my understanding, lies in two aspects: (i) **Frequency-Level Alignment**, which replaces high-frequency DCT coefficients of synthetic images with those from real images; and (ii) **Pixel-Level Alignment**, a mixup-style operation applied between real and synthetic images.
2. Regarding the first point, I have two main concerns. First, it is unclear whether the authors apply the DCT transform at the original image resolution or in 8×8 blocks—this is not explicitly specified in the paper. Second, the authors appear to define the high-frequency region as a bottom-right rectangle in the DCT space. However, DCT coefficients increase in frequency in a zigzag pattern from the top-left to the bottom-right. As such, this rectangular selection might omit some high-frequency components, making the design potentially suboptimal.
3. For the second point, while mixup is a well-established technique in image classification, it has not been applied in AIGI detection before, so I acknowledge this as a valid contribution. However, in Fig. 9(c), the ablation results show that even when rpixel=0rpixel=0 or rpixel=1rpixel=1, the detector still achieves high accuracy. This is counterintuitive—could the authors clarify why extreme values still perform well?
4. The detector is fine-tuned using DINOv2 with an input size of 336×336. It would be helpful for the authors to include ablations on different commonly used backbones (e.g., CLIP, ResNet-50) and input sizes (e.g., 224×224), since most baselines are trained under these settings. Also, any justification for selecting DINOv2 as the backbone would be appreciated.
5. The proposed method appears to be model-agnostic. Can the authors demonstrate whether DDA-aligned data could also enhance performance when used to train other existing detection methods?

**In summary**, this paper tackles a critical challenge in AIGI detection—robust generalization in real-world scenarios—and achieves state-of-the-art performance across multiple benchmarks. However, several technical and design questions remain to be addressed.



[1] Aligned Datasets Improve Detection of Latent Diffusion-Generated Images, ICLR'25

---

> ### Author Rebuttal · Authors · 2025-07-30
>
> Thank you for acknowledging **strong gneralization** and **efficiency** of our proposed DDA. We will alleviate your remaining concerns.
>
> ---
>
> > *Q1: Regarding the first point, I have two main concerns. First, it is unclear whether the authors apply the DCT transform at the original image resolution or in 8×8 blocks—this is not explicitly specified in the paper. Second, the authors appear to define the high-frequency region as a bottom-right rectangle in the DCT space. However, DCT coefficients increase in frequency in a zigzag pattern from the top-left to the bottom-right. As such, this rectangular selection might omit some high-frequency components, making the design potentially suboptimal.*
>
> Thank you for your pointting that concern.
>
> * **DCT Resolution:** We clarify that the DCT transform in our method is applied in 8×8 blocks, consistent with the standard JPEG compression pipeline. This design reflects a practical consideration: real images in most datasets (e.g., GenImage, ForenSynth) are JPEG-compressed, while many synthetic images are stored in PNG format. Applying block-wise DCT allows us to more effectively capture and mitigate compression-related frequency biases.
>
> * **High-Frequency Region Selection:** Below we conduct additional experiments using a **zigzag-pattern-based frequency selection**. As shown in Table 1 below, DDA with \$T\_{\text{freq}} = 0.2\$ (zigzag) achieves a comparable performance to \$T\_{\text{freq}} = 0.5\$ (rectangular). This suggests that our method is robust to the precise frequency indexing scheme. We will clarify both the DCT resolution and the frequency selection strategy in the revised manuscript.
>
> **Table 1. Ablation study of DDA using zigzag-based vs. rectangular high-frequency region selection.**
>
> | T_freq   | GenImage   | DRCT-2M   | DDA-COCO   | EvalGEN   | Synthbuster   | Chameleon   | SynthWildx   | Avg      |
> |:---------|:-----------|:----------|:-----------|:----------|:--------------|:------------|:-------------|:---------|
> | zigzag-0.1      | **96.4**   | 96.7      | 93.1       | 93.2      | *93.8*        | 72.5        | 82.2         | 89.7     |
> | zigzag-0.2      | **96.4**    | **98.5**  | 94.8       | 94.1      | 93.6          | *73.8*      | *83.1*       | **90.6** |
> | zigzag-0.3      | 94.0       | 95.8      | *94.9*     | 94.1      | 92.4          | 71.9        | 82.5         | 89.4     |
> | zigzag-0.4      | 92.3       | 93.3      | *94.9*     | *94.6*    | 92.6          | 72.5        | 80.8         | 88.7     |
> | zigzag-0.5      | 91.2       | 92.4      | **97.3**   | **95.1**  | 90.6          | 69.1        | 79.2         | 87.8     |
> | rectangular-0.5 | 95.5       | *97.4*    | 94.3       | 94.0      | **94.6**      | **74.3**    | **84.0**     | **90.6** |
>
> ---
>
> > *Q2: For the second point, while mixup is a well-established technique in image classification, it has not been applied in AIGI detection before, so I acknowledge this as a valid contribution. However, in Fig. 9(c), the ablation results show that even when rpixel=0 or rpixel=1, the detector still achieves high accuracy. This is counterintuitive—could the authors clarify why extreme values still perform well?*
>
> Thank you again for your careful reading and constructive feedback. We apologize for the confusion caused by the x-axis notation in Fig. 9 (c). The label should be **\$R\_{\text{pixel}}\$**, consistent with the figure caption. Specifically:
>
> * **\$R\_{\text{pixel}} = 0.0\$** means **no pixel-level mixup is applied** (i.e., frequency alignment only).
> * **\$R\_{\text{pixel}} = 1.0\$** means the pixel-level mixup ratio \$r\_{\text{pixel}}\$ is **sampled from a uniform distribution \$U\[0, 1]\$** during training (see Eq 3 of main paper).
>
> The strong performance at \$R\_{\text{pixel}} = 0\$ is not contradictory. It reflects the fact that **frequency-domain alignment alone is already highly effective**—achieving 91% accuracy in our ablation—since VAE reconstructions provide substantial low-level alignment. Similarly, the strong performance at **\$R\_{\text{pixel}} = 1.0\$** aligns with expectations. We will correct the axis label in Fig. 9 and add this clarification in the revision.
>
> ---
>
> > *Q3: The detector is fine-tuned using DINOv2 with an input size of 336×336. It would be helpful for the authors to include ablations on different commonly used backbones (e.g., CLIP, ResNet-50) and input sizes (e.g., 224×224), since most baselines are trained under these settings. Also, any justification for selecting DINOv2 as the backbone would be appreciated.*
>
> **A** Thank you for your comment. We clarify that **we have already provided ablation studies on both input sizes (Appendix Table 6) and backbone architectures (Appendix Table 7)**.
>
> * **Input Sizes:** Table 2 presents results for input resolutions of **224, 252, 280, 336, 392, 448, and 504**. The results show that DDA remains consistently effective across all tested resolutions.
>
> * **Backbones:** Table 3 compares DINOv2 and CLIP ViT-B/16. We observe that **DINOv2 outperforms CLIP**, likely due to its stronger focus on low-level, pixel-sensitive features that are more effective for capturing DDA-aligned artifacts. In contrast, CLIP is optimized for high-level semantics. We also attempted to train DDA with ResNet-50, but it failed to converge—likely due to insufficient representational capacity for modeling subtle DDA-induced artifacts.
>
> **Table 2. Ablation study of DDA across different input sizes.**
> | Input Size | GenImage | DRCT-2M  | DDA-COCO | EvalGEN  | Synthbuster | Chameleon | SynthWildx | Avg |
> |-------|--------|--------|----------|----------|-------|--------|---------|----------------|
> | 224   | 94.9   | 96.7   | **95.9** | **97.2** | 88.9 | 71.9   | 80.3    | 89.4 ± 9.8     |
> | 252   | 95.3   | 96.7   | 95.0     | 94.1     | 92.4  | 72.0   | 84.0    | 89.9 ± 8.9     |
> | 280   |**95.7**| 96.2   | *95.6*   | 95.4     | 91.9  | 70.1   | 84.6    | 89.9 ± 9.7     |
> | 392   | 92.9   | 96.5   | 92.0     | 95.7     | 93.9  | 71.8   | *89.6*  | *90.3 ± 8.5*   |
> | 448   | 93.4   | *97.2* | 90.7     | 89.5   |**95.8**| 65.7    |**89.9**| 88.9 ± 10.6    |
> | 504   | 93.0   | 93.0   | 92.7     | *95.8*   | 93.3  |*73.2*  | 86.2    | 89.6 ± 7.8     |
> |**336**| *95.5*|**97.4** | 94.3     | 94.0    | *94.6*|**74.3**| 84.0    | **90.6 ± 8.4** |
>
>
> **Table 3. Ablation study of DDA across different backbones.**
> | Method    | Backbone            | GenImage | DRCT-2M  | DDA-COCO | EvalGEN  | Synthbuster | Chameleon | SynthWildx | Avg            |
> | --------- |-------------------- | -------- | -------- | -------- | -------- | ----------- | --------- | ---------- | -------------- |
> | Fatformer |CLIP ViT-L/14        | 62.8     | 52.2     | 49.9    | 45.6     | 56.1        | 51.2      | 52.1       | 52.8 ± 5.4     |
> | UnivFD    |CLIP ViT-L/14        | 64.1     | 61.8     | 51.4     | 15.4     | 67.8        | 50.7      | 52.3       | 51.9 ± 17.5    |
> | DRCT   |CLIP ViT-L/14        | 84.7     | *90.5*     | 62.3     | 77.7     | *84.8*      | 56.6      | 55.1       | 73.1 ± 14.8    |
> | C2P-CLIP |CLIP ViT-L/14        | 74.4     | 59.2     | 49.9     | 38.9     | 68.5        | 51.1      | 57.1       | 57.0 ± 11.9    |
> | AIDE      |CLIP ConvNeXt| 61.2| 64.6  | 50.0     | 15.0     | 53.9        | 63.1      | 48.8       | 50.9 ± 17.1    |
> | DDA       |CLIP ViT-B/16        | 95.2     | 80.3     | *97.9*   | *96.2*     | 55.5        | 46.6      | 62.0       | 76.2 ± 21.4    |
> | DDA       |CLIP ViT-L/14        | **97.0** | 80.4     | **98.8**| **99.2** | 68.3        | *67.7*    | *71.8*     | *83.3 ± 14.7*  |
> | DDA       |**DINOv2 VIT-L/14** | *95.5*   | **97.4** | 94.3    | 94.0     | **94.6**     | **74.3** | **84.0**   | **90.6 ± 8.4** |
>
> ---
>
> > *Q4: The proposed method appears to be model-agnostic. Can the authors demonstrate whether DDA-aligned data could also enhance performance when used to train other existing detection methods?*
>
> Thanks for your insightful comment. To assess whether DDA-aligned data benefits existing detection models, we conducted additional experiments that **isolate the effect of DDA**. Specifically, we replaced the synthetic training images in baseline methods with **DDA-aligned counterparts**, while keeping all other components—including model architecture, training settings, and loss functions—unchanged. The results, summarized in **Table 4**, show **consistent and significant improvements in accuracy**, confirming that DDA-aligned data enhances generalization
>
> **Table 4. Evluation of baseline methods with and without DDA-aligned synthetic data.**
>
> | Method          | GenImage         | DRCT-2M          | DDA-COCO         | EvalGEN          | Chameleon        | WildRF          |  AVG             |
> |-----------------|------------------|------------------|------------------|------------------|------------------|-----------------|------------------|
> | UnivFD       |  64.1            | 61.8             | 51.4             | 15.4             | 50.7             | 55.3            | 49.8             |
> | UnivFD + DDA    |**92.4 (↑28.3)** | **76.1 (↑14.3)** | **78.2 (↑26.8)**| **98.7 (↑83.3)**| **65.6 (↑14.9)**| **56.6 (↑1.3)** |**77.9 (↑28.1)** |
> | Fatformer   |  62.8            | 52.2             | 49.9             | 45.6             | 51.2             | 58.9            | 53.4             |
> | Fatformer + DDA | **65.5 (↑2.7)** | **58.9 (↑6.7)**  | **68.6 (↑18.7)**| **77.0 (↑31.4)**| **54.0 (↑2.8)** | 51.3 (↓7.6)      | **62.6 (↑9.2)** |
> | DRCT        |  84.7            | 90.5             | 62.3             | 77.7             | 56.6             | 50.6            | 70.4             |
> | DRCT + DDA      | **91.7 (↑7.0)** | 86.2 (↓4.3)       |**77.3 (↑15.0)** | **97.2 (↑19.5)**| **68.0 (↑11.4)**| **54.8 (↑4.2)** | **79.2 (↑8.8)** |

---

> ### Comment · Reviewer_Two9 · 2025-08-01
>
> I have read the author's response, and most of the concerns have been addressed. The only remaining concern is the issue raised in Q1 regarding High-Frequency Region Selection. Although the `rectangular-0.5` setting shows performance similar to the `zigzag` method, this choice is not theoretically convincing, as it does not align with the high-frequency distribution of the DCT and may miss some compressed high-frequency components.
>
> I hope the authors will provide a more in-depth discussion or revise the corresponding method in the revised version. Compared to the overall contribution of this paper, this minor concern does not overshadow its merits. I am willing to update my rating to `accept` after discussion. Good luck!

---

> > ### Author Response · Authors · 2025-08-03
> > **Official Comment by Authors**
> >
> > Thank you so much for your positive feedback! It encourages us a lot.
> >
> > We are glad our responses have addressed your concerns and appreciate your willingness to recommend acceptance after discussion! Your suggestion on high‑frequency region selection is valuable; we will refine the discussion and add further comparisons in the revision.
> >
> > We sincerely thank you for your thoughtful comments and time, which have been essential in improving the quality of our work.

---

### Official Review · Reviewer_ks8s · 2025-07-02

**Clarity:** 4
**Significance:** 3
**Originality:** 3
**Rating:** 5
**Confidence:** 4

**Summary:**

This paper proposes a new method for AI-generated Image (AIGI) detection. The paper hypothesizes that dataset bias, both in pixel-level semantics and in frequency domain is an important factor for the insufficient generalizability of the existing detection methods to unseen generative models. Therefore, this paper, for the first time introduces a dual data alignment (DDA) pipeline to minimize the effect of dataset bias and create a real vs. AI-generated dataset from MS-COCO (referred to as DDA-COCO).  By fine-tuning a DINOv2 backbone using LoRA on DDA-COCO, the proposed method achieves state-of-the-art performance in terms of balanced accuracy on several existing benchmarks as well as two new benchmarks introduces in this paper.

**Questions:**

Referring to the weaknesses section I have the following questions:

1. How would existing methods' performance change if DDA was used to minimize the training dataset bias?
2. How AP or AUROC of the proposed method compare with that of the existing methods?
3. How thresholds for different methods are chosen in this study?

Based on the authors' response, I would be willing to increase my rating to 5 (accept).

**Ethical Concerns:**

["NO or VERY MINOR ethics concerns only"]

**Final Justification:**

All of my concerns are addressed in the authors' rebuttal, and I feel more confident in accepting the paper. Therefore, I raise my initial rating to 5.

**Limitations:**

Yes

**Paper Formatting Concerns:**

No major formatting concerns were identified.

**Quality:**

3

**Strengths And Weaknesses:**

**Strengths**
1. The paper is well-written, and it is easy to follow.
2. The frequent use of clear visualizations and figures helps with understanding the concepts and the proposed novelties.
3. The proposed method is properly motivated and is an easy-to-understand and elegant solution.
4. The proposed method shows robust performance across several datasets, achieving state-of-the-art results.
5. The experimental results are extensive and the proposed method's performance is compared against recently published papers in top-tier venues on recent benchmarks.

**Weaknesses**
1. The main hypothesis of the paper (dataset bias is the problem and that the data alignment helps) is not verified in isolation. Although extensive experimental results show strong performance of the proposed method, the strong performance cannot be solely attributed to the use of the data alignment pipeline. There are several differences between the proposed method and existing approaches other than the use of DDA. For example, the original source of data used in this paper is different from those used in competitive methods. Additionally, the backbone, the fine-tuning strategy and the input size to the backbone are all different compared to existing methods. Given this, it is hard to be sure if DDA is the main reason behind the proposed method's strong performance.
2. The paper does not report threshold-less metrics such as AP or AUROC which are commonly used in many published papers in this area. Threshold-less metrics are important measures of the separability between the representation of real vs AI-generated samples. Additionally, the difference in accuracy numbers can be attributed to the poor choice of the decision thresholds.
3. The mechanism for choosing a decision threshold is not discussed in the paper.

---

> ### Author Rebuttal · Authors · 2025-07-30
>
> We are grateful for your positive recognition of our **novelty, extensive experiments** and **writing**! We will alleviate your remaining concerns.
>
> ---
>
> > *Q1: The main hypothesis of the paper (dataset bias is the problem and that the data alignment helps) is not verified in isolation. Although extensive experimental results show strong performance of the proposed method, the strong performance cannot be solely attributed to the use of the data alignment pipeline. There are several differences between the proposed method and existing approaches other than the use of DDA. For example, the original source of data used in this paper is different from those used in competitive methods. How would existing methods' performance change if DDA was used to minimize the training dataset bias?*
>
> Thank you for this thoughtful question.
>
> We respectfully clarify that, in line with established evaluation practices, we use the **official checkpoints released by the original authors** for baseline methods—**a standard protocol also followed in prior works such as FatFormer, C2P-CLIP, AIDE, and DRCT**—ensuring consistency and fairness in comparison.
>
> To directly address your concern, we conduct a **controlled one-to-one comparison**, where we adopt the **same architecture, training strategy, and real image source as the competitive method**, but **replace its synthetic images of its training set with DDA-aligned images**. This setup isolates the impact of DDA while keeping all other factors constant. Preliminary results show clear performance improvements.
>
> **Table 1. Controlled one-to-one comparison of existing methods with and without DDA-aligned training data.** Each baseline method (UnivFD, FatFormer, DRCT) is retrained under identical settings, replacing only the original synthetic training data with DDA-aligned samples. DDA consistently improves performance across all datasets.
>
> | Method          | GenImage         | DRCT-2M          | DDA-COCO         | EvalGEN          | Chameleon        | WildRF          |  AVG             |
> |-----------------|------------------|------------------|------------------|------------------|------------------|-----------------|------------------|
> | UnivFD       |  64.1            | 61.8              | 51.4            | 15.4             | 50.7             | 55.3            | 49.8             |
> | UnivFD + DDA |**92.4 (↑28.3)** | **76.1 (↑14.3)** | **78.2 (↑26.8)**| **98.7 (↑83.3)**| **65.6 (↑14.9)**| **56.6 (↑1.3)** |**77.9 (↑28.1)** |
> | Fatformer       |  62.8            | 52.2             | 49.9             | 45.6             | 51.2             | 58.9            | 53.4             |
> | Fatformer + DDA | **65.5 (↑2.7)** | **58.9 (↑6.7)**  | **68.6 (↑18.7)**| **77.0 (↑31.4)**| **54.0 (↑2.8)** | 51.3 (↓7.6)      | **62.6 (↑9.2)** |
> | DRCT        |  84.7            | 90.5             | 62.3             | 77.7             | 56.6             | 50.6            | 70.4             |
> | DRCT + DDA  | **91.7 (↑7.0)** | 86.2 (↓4.3)       |**77.3 (↑15.0)** | **97.2 (↑19.5)**| **68.0 (↑11.4)**| **54.8 (↑4.2)** | **79.2 (↑8.8)** |
>
>
> > *Q2: The paper does not report threshold-less metrics such as AP or AUROC which are commonly used in many published papers in this area. Threshold-less metrics are important measures of the separability between the representation of real vs AI-generated samples.*
>
> Thank you for this thoughtful suggestion regarding threshold-independent evaluation metrics. We clarify that, in line with prior works such as C2P-CLIP, DRCT, AlignedForensics, and AIDE, our main paper reports balanced accuracy for comparability. Following your suggestion, we have additionally computed **AP and AUROC scores** for our method. Our method **DDA achieves state-of-the-art performance**, with average scores of **0.964(AP)** and **0.967(AUROC)**—outperforming all baselines by a non-trivial margin. These results confirm the superior performance of DDA.
>
> **Table 2. Overall Comparison of AP / AUROC.** Bold numbers indicate the best score per row; values in parentheses denote the absolute improvement over the original method.
>
> | Method | DRCT-2M | GenImage | Synthbuster | SynthWildx | WildRF | AIGCDetection  Benchmark | ForenSynth | Chameleon | AVG | MIN |
> |--------|----------|----------|----------|----------|----------|----------|----------|----------|----------|----------|
> | NPR (CVPR'24) | 0.403/0.271 | 0.501/0.440 | 0.509/0.515 | 0.529/0.533 | 0.742/0.702 | 0.464/0.372 | 0.450/0.338 | 0.517/0.551 | 0.514/0.465 | 0.403/0.271 |
> | UnivFD (CVPR'23) | 0.857/0.864 | 0.825/0.838 | 0.792/0.797 | 0.521/0.463 | 0.624/0.541 | 0.868/0.879 | 0.918/0.921 | 0.477/0.554 | 0.735/0.732 | 0.477/0.463 |
> | FatFormer (CVPR'24) | 0.478/0.386 | 0.715/0.684 | 0.580/0.560 | 0.572/0.584 | 0.759/0.707 | 0.920/0.907 | *0.981/0.975* | 0.614/0.608 | 0.702/0.676 | 0.478/0.386 |
> | SAFE (KDD'25) | 0.577/0.554 | 0.539/0.554 | 0.542/0.527 | 0.496/0.491 | 0.707/0.621 | 0.520/0.524 | 0.542/0.545 | 0.506/0.571 | 0.554/0.548 | 0.496/0.491 |
> | C2P-CLIP (AAAI'25) | 0.707/0.652 | 0.923/0.909 | 0.876/0.859 | 0.671/0.685 | 0.751/0.727 | *0.933/0.921* | **0.982/0.978** | 0.464/0.442 | 0.788/0.772 | 0.464/0.442 |
> | AIDE (ICLR'25) | 0.702/0.705 | 0.755/0.767 | 0.499/0.448 | 0.466/0.438 | 0.714/0.647 | 0.792/0.806 | 0.768/0.740 | 0.430/0.454 | 0.641/0.626 | 0.430/0.438 |
> | DRCT (ICML'24) | *0.961/0.965* | *0.939/0.949* | *0.901/0.903* | 0.576/0.598 | 0.595/0.534 | 0.907/0.917 | 0.890/0.898 | 0.663/0.719 | 0.804/0.810 | 0.576/0.534 |
> | AlignedForensics (ICLR'25) | 0.998/0.998 | 0.930/0.947 | 0.796/0.805 | *0.870/0.849* | *0.905/0.854* | 0.807/0.798 | 0.670/0.650 | **0.835/0.854** | *0.851/0.844* | *0.670/0.650* |
> | DDA (ours) | **0.998/0.998** | **0.990/0.991** | **0.992/0.993** | **0.972/0.971** | **0.982/0.981** | **0.989/0.990** | 0.969/0.972 | *0.824/0.841* | **0.965/0.967** | **0.824/0.841** |
>
>
> > *Q3: The mechanism for choosing a decision threshold is not discussed in the paper.*
>
> Thank you for pointing this out. We clarify that our DDA-based binary classifier uses **a fixed decision threshold of 0.5**: samples with predicted logits greater than 0.5 are classified as synthetic, and those below as real. No threshold tuning or calibration is applied during evaluation.

---

> > ### Comment · Reviewer_ks8s · 2025-08-06
> > **Strong rebuttal**
> >
> > I thank the authors for their rebuttal and addressing my concerns. I think it would be very helpful to add these new results and clarifications to the paper or the supplementary material, especially Table 1 in the rebuttal. All of my concerns are addressed, and I feel more confident in accepting the paper. Therefore, I raise my initial rating to 5.

---

> > > ### Author Response · Authors · 2025-08-07
> > >
> > > **Dear Reviewer,**
> > >
> > > Thank you for your thoughtful feedback and for raising the rating! We greatly appreciate your suggestions and will be sure to incorporate these updates into the revised manuscript.
> > >
> > > Best regards,
> > >
> > > Authors

---

> ### Author Response · Authors · 2025-08-06
>
> **Dear Reviewer,**
>
> Thank you again for your valuable efforts and constructive advice in reviewing our paper. **As the discussion period nears its end, we look forward to your feedback on our responses.** We have made every effort to address all your concerns and are happy to clarify any points or discuss any remaining questions.
>
> Best regards,
>
> Authors

---

### Official Review · Reviewer_sHpt · 2025-07-05

**Clarity:** 3
**Significance:** 3
**Originality:** 3
**Rating:** 3
**Confidence:** 5

**Summary:**

This study identifies that single reconstruction alone does not suffice to achieve comprehensive alignment between real and synthetic image pairs. To address this limitation, the authors propose a novel approach termed Dual Data Alignment (DDA), which aligns synthetic images with their real counterparts in both pixel and frequency domains, thereby reducing bias in AIGI detectors. Furthermore, two new AIGI datasets are presented to expand testing scenarios across diverse domains. Extensive evaluations conducted on eight benchmark datasets validate the effectiveness of the proposed methodology.

**Questions:**

1. Please refer to weakness.
2. The authors should clarify the fairness of the experimental comparisons. Specifically, in Table 3, it is unclear whether the proposed method and the baseline methods were trained on the same datasets and under comparable settings. This information is critical for a fair evaluation.
3. The backbone used in the proposed method is DINOv2, while some baselines adopt different backbones such as CLIP or ResNet. The authors should discuss how the choice of backbone influences performance and whether it contributes significantly to the observed improvements.

**Ethical Concerns:**

["NO or VERY MINOR ethics concerns only"]

**Final Justification:**

I would like to thank the authors for their rebuttal.
In their response, some of my doubts were addressed. However, I still have concerns about fair comparisons. The authors claim to have used the official checkpoints of the baselines, but different training data were used for different baselines in the table, which makes the comparison unfair. Overall, the dataset proposed by the authors is interesting, but the experimental comparisons are concerning. I keep my score.

**Limitations:**

The paper should provide a more comprehensive discussion on the motivations behind image reconstruction as well as address issues related to experimental fairness, ensuring greater transparency and equity in the evaluation process.

**Paper Formatting Concerns:**

The  NPR is presented in a paper at CVPR 2024.

**Quality:**

3

**Strengths And Weaknesses:**

Strengths:
1. The writing of this manuscript is generally clear and easy to follow.
2. The topic of detecting AIGC (AI-Generated Content) images is both timely and interesting.
3. The authors have conducted extensive experiments, which demonstrate the effectiveness and relevance of the proposed method.
4. The proposed method is novel in that it simultaneously considers both pixel-level and frequency-domain alignment.

Weaknesses：
1. The authors claim that existing datasets suffer from biases in format, content, and size. However, these biases can often be mitigated through data augmentation or by expanding the dataset, without the need for complex reconstruction-based approaches. For instance, JPEG compression and cropping augmentation could address format and size biases effectively.
2. The primary objective of reconstruction-based methods is typically to uncover the intrinsic differences between real and fake images, rather than to align real and fake data distributions.
3. It is unclear whether the proposed method and the baselines in Table 3/4/5/6 were trained under the same conditions, particularly with respect to the training dataset. Clarification on this point would be helpful for a fair comparison.
4. Since the proposed method uses DINOv2 as the backbone, whereas some baselines rely on CLIP or ResNet, it would be important to discuss how the choice of backbone affects the results. This would help ensure a fair and meaningful comparison.
5. The proposed method has not been evaluated on the ForenSynths dataset (CNNSpot CVPR 2020), a commonly used benchmark for detecting CNN-generated images. This limits the completeness of the experimental validation.
6. The generalization capability of the proposed DDA method to GAN-generated images is not discussed. It would be beneficial to evaluate its performance on such images to better understand its applicability across different types of generative models.

---

> ### Author Rebuttal · Authors · 2025-07-31
>
> We appreciate your positive comments on our **novelty, extensive experiments** and **writing**! We will alleviate your remaining concerns.
>
> ---
>
> > *Q1: The biases in format, content, and size can be mitigated through data augmentation or by expanding the dataset, without the need for reconstruction.*
>
> Thank you for raising this important concern.
>
> **Content and size bias:** reconstruction-based methods able to generate aligned synthetic counterparts, preserving content while altering only generation-specific characteristics. in contrast, dataset expansion cannot guarantee precise semantic alignment in every detail (e.g., object types, textures, and layouts), leaving content bias unaddressed.
>
> **On the complexity of reconstruction-based approaches:** We respectfully disagree that reconstruction-based methods are overly complex. With modern frameworks (e.g., *diffusers*), VAE reconstruction is accessible and computationally lightweight. Table 10 of our main paper shows DDA is more efficient than many existing baselines in terms of generation time.
>
> **Format bias:** Due to the asymmetric encoding in real vs. synthetic training images (JPEG-compressed real vs. PNG synthetic), JPEG augmentation can result in **double-compressed real images** versus **single-compressed synthetic images**. Consequently, models may learn to associate stronger compression artifacts with authenticity. We empirically substantiate this in Tables 1 and 2:
>
> * **Table 1** shows that VAE reconstruction + JPEG augmentation exhibits **a significant drop (↓22.0)** when tested on JPEG-format synthetic images, indicating format bias. In contrast, DDA maintains stable performance (↑3.0).
>
> * **Table 2** evaluates the frequency-based detector SAFE. Even with JPEG augmentation, SAFE suffers a **a significant drop (↓21.0)** in accuracy on JPEG-format images. This highlights that augmentation-only methods fail to completely eliminate format bias.
>
> **Table 1. Evaluation of JPEG compression augmentation for mitigating format bias.** VAE reconstruction with JPEG compression augmentation (VAE Rec. + JPEG Aug) versus VAE reconstruction with our proposed Dual Data Alignment (VAE Rec. + DDA). We report accuracies on detecting PNG-format and JPEG-format synthetic images on GenImage.
>
> | Method            | Format | Midjourney |SD14| SD15| ADM  | GLIDE|Wukong| VQDM |BigGAN| AVG ± STD           |
> |-------------------|--------|-------|--------|------|------|------|------|------|------|---------------------|
> | VAE Rec. + JPEG Aug | PNG  |  86.5 |  100.0 | 99.8 | 86.0 | 86.5 | 99.9 | 91.3 | 68.9 | 89.9 ± 10.6         |
> | VAE Rec. + JPEG Aug | JPG  |  92.2 |   98.9 | 98.9 | 45.2 | 67.3 | 99.2 | 40.2 | 1.3  | 67.9 ± 36.3 (↓22.0) |
> | VAE Rec. + DDA      | PNG  |  93.5 |   99.7 | 99.5 | 86.0 | 84.2 | 99.5 | 89.5 | 93.6 | 93.2 ± 6.2          |
> | VAE Rec. + DDA      | JPG  |  94.3 |   99.9 | 99.6 | 93.6 | 91.0 | 99.7 | 94.1 | 97.1 | 96.2 ± 3.4 (↑3.0)   |
>
>
> **Table 2. Evaluation of format bias mitigation for SAFE.**
>
> | Method          | Format| Midjourney | SD14 | SD15 | ADM  | GLIDE | Wukong | VQDM | BigGAN | AVG ± STD           |
> |-----------------|-------|------------|------|------|------|-------|--------|------|--------|---------------------|
> | SAFE            | PNG   |       91.2 | 99.5 | 99.4 | 64.7 |  93.3 |   97.2 | 93.3 |   96.6 | 91.9 ± 11.4         |
> | SAFE            | JPG   |        0.5 |  1.7 | 2.0  |  1.5 |   8.2 |  3.0   |  2.7 |    4.6 | 3.0 ± 2.4 (↓88.9)   |
> | SAFE + JPEG Aug | PNG   |       90.3 | 96.8 | 96.3 | 62.2 |  91.9 |   89.7 | 73.1 |   89.2 | 86.2 ± 12.1         |
> | SAFE + JPEG Aug | JPG   |       60.7 | 61.5 | 61.0 | 80.6 |  83.1 |   63.3 | 76.2 |   35.1 | 65.2 ± 15.3 (↓21.0) |
>
> ---
>
> > *Q2: Unclear whether the proposed method and the baselines in Table 3/4/5/6 were trained under the same conditions.*
>
> Thank you for raising this important concern.
>
> **Clarification on training conditions:** We respectfully clarify that for the comparisons in Tables 3–6, we follow established practices by using the **official checkpoints released by the original authors** for all baseline methods. This evaluation protocol is also adopted in prior work such as AIDE, and DRCT.
>
> * **Fair comparison:** We acknowledge that DDA may benefit from certain training setups in Table 3. To provide a more comprehensive and balanced view, we include an extended evaluation in **Appendix Table 1** (see following Table 3). **DDA achieves SoTA on 9 out of 10 datasets.**
>
> * **One-to-one comparisons:** In Table 4 we conduct additional experiments where all training variables are held constant, and the only change is substituting the synthetic training data with **DDA-aligned counterparts**. These controlled results consistently show that **DDA significantly enhances generalization performance**, isolating the impact of our alignment strategy.
>
> **Table 3: Comprehensive evaluation of DDA against state-of-the-art detectors on 10 benchmark datasets comprising 561k images from 12 GANs, 52 diffusion models, and 2 autoregressive models, including 3 in-the-wild datasets.**
>
> | Method                     | GenImage | DRCT-2M  | DDA-COCO | EvalGEN  | Synthbuster | ForenSynth | AIGCDetection  Benchmark | Chameleon | Synthwildx | WildRF   | Avg            | Min      |
> | -------------------------- | -------- | -------- | -------- | -------- | ----------- |---------- | ----------------------- | --------- | ---------- | -------- | -------------- | -------- |
> | NPR (CVPR'24)              | 51.5     | 37.3     | 28.1     | 59.2     | 50.0    | 47.9    | 53.1     | 59.9      | 49.8       | 63.5     | 50.0 ± 10.7    | 28.1     |
> | UnivFD (CVPR'23)           | 64.1     | 61.8     | 3.6      | 15.4     | 67.8    | 77.7    | 72.5     | 50.7      | 52.3       | 55.3     | 52.1 ± 24.2    | 3.6      |
> | FatFormer (CVPR'24)        | 62.8     | 52.2     | 3.3      | 45.6     | 56.1    | *90.1*  | *85.0*   | 51.2      | 52.1       | 58.9     | 55.7 ± 23.6    | 3.3      |
> | SAFE (KDD'25)              | 50.3     | 59.3     | 0.5      | 1.1      | 46.5    | 49.7    | 50.3     | 59.2      | 49.1       | 57.2     | 42.3 ± 22.3    | 0.5      |
> | C2P-CLIP (AAAI'25)         | 74.4     | 59.2     | 2.0      | 38.9     | 68.5    |**92.1** | 81.4     | 51.1      | 57.1       | 59.6     | 58.4 ± 25.0    | 2.0      |
> | AIDE (ICLR'25)             | 61.2     | 64.6     | 1.2      | 15.0     | 53.9    | 59.4    | 63.6     | 63.1      | 48.8       | 58.4     | 48.9 ± 22.3    | 1.2      |
> | DRCT (ICML'24)             | *84.7*   | 90.5     | 30.4     | *77.7*   | *84.8*  | 73.9    | 81.4     | 56.6      | 55.1       | 50.6     | 68.6 ± 19.4    | 30.4     |
> | AlignedForensics (ICLR'25) | 79.0     | *95.5*   | *86.6*   | 77.0     | 77.4    | 53.9    | 66.6     | *71.0*    | *78.8*     | *80.1*   | *76.6* ± 11.2  | *53.9*   |
> | **DDA (ours)**            | **95.5** | **97.4**| **94.3** | **94.0** |**94.6**| 85.5     | **93.3**| **74.3**  | **84.0**   | **95.1** | **90.8** ± 7.3 | **74.3** |
>
>
> **Table 4. One-to-One comparisons.**
>
> | Method          | GenImage         | DRCT-2M          | DDA-COCO         | EvalGEN          | Chameleon        | WildRF          |  AVG             |
> |-----------------|------------------|------------------|------------------|------------------|------------------|-----------------|------------------|
> | UnivFD       |  64.1            | 61.8             | 51.4             | 15.4             | 50.7             | 55.3            | 49.8             |
> | UnivFD + DDA    |**92.4 (↑28.3)** | **76.1 (↑14.3)** | **78.2 (↑26.8)**| **98.7 (↑83.3)**| **65.6 (↑14.9)**| **56.6 (↑1.3)** |**77.9 (↑28.1)** |
> | Fatformer   |  62.8            | 52.2             | 49.9             | 45.6             | 51.2             | 58.9            | 53.4             |
> | Fatformer + DDA | **65.5 (↑2.7)** | **58.9 (↑6.7)**  | **68.6 (↑18.7)**| **77.0 (↑31.4)**| **54.0 (↑2.8)** | 51.3 (↓7.6)      | **62.6 (↑9.2)** |
> | DRCT        |  84.7            | 90.5             | 62.3             | 77.7             | 56.6             | 50.6            | 70.4             |
> | DRCT + DDA      | **91.7 (↑7.0)** | 86.2 (↓4.3)       |**77.3 (↑15.0)** | **97.2 (↑19.5)**| **68.0 (↑11.4)**| **54.8 (↑4.2)** | **79.2 (↑8.8)** |
>
> ---
>
> > *Q3: The impact of backbone.*
>
> Thank you for this question. We respectfully point out that **we have already conducted ablation studies on backbones in Appendix Table 7.** Below, we provide a simplified version of the results. While **DDA performs best with DINOv2**, it still **significantly outperforms all baseline methods when using CLIP**.
>
> **Table 5. Ablation study on backbone.**
>
> | Method    |Backbone             | GenImage | DRCT-2M |DDA-COCO| EvalGEN |Synthbuster|Chameleon |SynthWildx | Avg         |
> | --------- |---------------------|----------|---------|--------|---------|-----------|----------|-----------|-------------|
> | Fatformer |CLIP ViT-L/14        | 62.8      | 52.2   | 49.85   | 45.6    | 56.1     | 51.2     | 52.1      | 52.8 ± 5.4  |
> | UnivFD    |CLIP ViT-L/14        | 64.1     | 61.8    | 51.4   | 15.4    | 67.8      | 50.7     | 52.3      | 51.9 ± 17.5 |
> | DRCT      |CLIP ViT-L/14        | 84.7     | 90.5    | 62.3   | 77.7    | *84.8*    | 56.6     | 55.1      | 73.1 ± 14.8 |
> | C2P-CLIP  |CLIP ViT-L/14        | 74.4     | 59.2    | 49.9   | 38.9    | 68.5      | 51.1     | 57.1      | 57.0 ± 11.9 |
> | AIDE   |CLIP-ConvNeXt | 61.2| 64.6 | 50.0   | 15.0    | 53.9      | 63.1     | 48.8      | 50.9 ± 17.1 |
> | DDA       |CLIP ViT-L/14        | **97.0** | 80.4    |**98.8**|**99.2**| 68.3      | *67.7*   | *71.8*    |*83.3 ± 14.7*|
> | DDA       |DINOv2 VIT-L/14      | *95.5*   | **97.4**| 94.3   | 94.0    | **94.6** | **74.3** | **84.0**  |**90.6 ± 8.4**|
>
> ---
>
> > *Q4: The proposed method has not been evaluated on ForenSynths.*
>
> Thank you for raising this important concern. We respectfully clarify that **DDA has been already evaluated on ForenSynth in Appendix Table 3**.

---

### Official Review · Reviewer_xaYE · 2025-07-07

**Clarity:** 3
**Significance:** 4
**Originality:** 4
**Rating:** 5
**Confidence:** 5

**Summary:**

This paper introduces Dual Data Alignment (DDA), a method to improve the generalizability of AI-generated image (AIGI) detectors by addressing dataset biases. Existing detectors struggle with new data because they often overfit on non-causal attributes like image format or size. DDA aligns synthetic and real images in both pixel and frequency domains, a crucial improvement over pixel-level alignment alone, which still leaves frequency-level discrepancies. The method involves VAE reconstruction, high-frequency fusion, and pixel mixup. DDA demonstrates significant performance improvements across diverse benchmarks, including new datasets like DDA-COCO and EvalGEN, highlighting its ability to create more unbiased and robust AIGI detectors

**Questions:**

1. Clarify and Substantiate the "Universal Upsampling Artifact" Claim.
Question: The paper states: "We hypothesize that this artifact arises during the VAE-based decoding process". While interesting, this remains a hypothesis. What specific characteristics define this "universal upsampling artifact"? Can the authors provide more empirical evidence or theoretical reasoning to support its "universality" across diverse generative models beyond the VAE decoding stage?

2. Address Performance on Heavily Post-Processed Images (Chameleon Dataset).
Question: The paper acknowledges that "Our method performs relatively lower on FLUX... and struggles to detect images with strong post-processing artifacts, as shown in the results on the Chameleon dataset". Given that real-world scenarios often involve aggressive post-processing, how do the authors envision mitigating this limitation? Is DDA inherently sensitive to certain types of artifacts, or are there planned extensions to improve robustness in these challenging cases?

3. Re-evaluate "Theory Assumptions and Proofs" Claim for Clarity.
Question: The NeurIPS checklist response states "Yes" for "For each theoretical result, does the paper provide the full set of assumptions and a complete (and correct) proof?" with a justification referring to Section 3.2. However, Section 3.2 primarily describes the methodology and provides intuitive explanations, rather than formal theoretical results, theorems, or proofs. This might be a misunderstanding of what constitutes a "theoretical result" in this context.

**Ethical Concerns:**

["NO or VERY MINOR ethics concerns only"]

**Final Justification:**

Accepted.

**Limitations:**

Yes

**Quality:**

3

**Strengths And Weaknesses:**

Strengths:
Quality based on robust experimental validation - The paper presents extensive evaluations across eight diverse datasets, including "in-the-wild" benchmarks, which significantly strengthens its claims of improved generalizability. The consistent outperformance of DDA over state-of-the-art methods (e.g., +12.4% on GenImage, +9.8% on Synthbuster, +17.7% on EvalGEN) is a strong indicator of its effectiveness. The inclusion of new, challenging test sets like DDA-COCO and EvalGEN is particularly valuable for rigorously assessing detector performance against new generative architectures and aligned data. The robustness analysis under various post-processing methods (JPEG compression, resizing, blurring) further validates DDA's practical utility, showing its ability to maintain high performance even when images are altered. This is critical for real-world deployment where images are frequently compressed or modified. The paper tackles the fundamental issue of dataset bias in AIGI detection, a well-recognized challenge that hinders the real-world applicability of detectors. By identifying and addressing both pixel-level and, crucially, frequency-level misalignment, the authors pinpoint a subtle yet significant source of bias that previous reconstruction methods missed.

Clarity - The paper clearly articulates the "frequency-level misalignment" problem with existing reconstruction methods, using Figure 3 and Figure 4 to visually and empirically demonstrate the issue. This strong motivation for DDA's dual-domain alignment is a major plus. The paper addresses very clear methodology, Figure 5 provides a clear visual pipeline of the proposed method.

Significance - The primary significance lies in the demonstrated improvement in generalizability for AIGI detectors. This is a crucial step towards more reliable and deployable fake image detection systems, addressing a pressing societal concern regarding misinformation and fraud.

Weaknesses:
Quality - The paper Limited Explanation of T freq and R pixel Parameter Selection and needs more information.

Clarity - "Theory Assumptions and Proofs" Claim: The paper states "Yes" for providing full assumptions and complete/correct proofs in Section 3.2. However, Section 3.2 primarily describes the methodology and motivation, not formal theorems, lemmas, or rigorous mathematical proofs in the traditional sense. This checklist answer might be misleading and could be interpreted as a claim of theoretical contribution that isn't fully supported by the content of Section 3.2.

---

> ### Author Rebuttal · Authors · 2025-07-29
>
> We sincerely thank you for your valuable time and comments. We are encouraged by your positive comments on the **significance** and **robust experimental validation** of our work! We will alleviate your remaining concerns as follows.
>
> ---
>
> > *Q1: The paper Limited Explanation of $T_{freq}$ and $R_{pixel}$ Parameter Selection and needs more information.*
>
> Thank you for the helpful suggestion. We clarify that \$T\_{\text{freq}}\$ and \$R\_{\text{pixel}}\$ control the degree of alignment in the frequency and pixel domains, respectively. Increasing their values enhances alignment strength, which can improve sensitivity to subtle generative artifacts. However, excessively strong alignment may shift the decision boundary too close to real images, potentially reducing true-positive accuracy. To balance this trade-off, we empirically set \$T\_{\text{freq}} = 0.5\$ and \$R\_{\text{pixel}} = 0.8\$, based on comprehensive validation in Figure 9 of the main paper.
>
> ---
>
> > *Q2: Clarify and Substantiate the "Universal Upsampling Artifact" Claim. Question: The paper states: "We hypothesize that this artifact arises during the VAE-based decoding process". While interesting, this remains a hypothesis. What specific characteristics define this "universal upsampling artifact"? Can the authors provide more empirical evidence or theoretical reasoning to support its "universality" across diverse generative models beyond the VAE decoding stage?*
>
> Thank you for this insightful question.
>
> **• What the universal artifact is:** We believe that the universal upsampling artifact stems from deterministic local correlations introduced by fixed, low-rank upsampling operations (e.g., bilinear interpolation, transposed convolution). These components, widely used in the decoders of VAEs, GANs, and diffusion models, project low-dimensional latents to high-resolution outputs. However, due to their limited representational capacity, they cannot fully capture the complexity of natural image statistics. As a result, generated images often exhibit reduced local rank and unnatural pixel dependencies—properties rarely seen in real images. These artifacts are therefore architectural in origin, rather than model-specific. Similar observations have been made in prior works such as NPR [1] and SPSL [2],
>
> **• Empirical evidence for university beyond the VAE decoding:** To substantiate universality, we highlight the results in Appendix Table 1. Our DDA detector, trained *only* on aligned VAE-reconstructed images, achieves SoTA on 9 out of 10 benchmarks. This strong generalization suggests that the universality is not confined to a specific generation mechanism.
>
> **Table 1: Comprehensive evaluation of DDA against state-of-the-art detectors on 10 benchmark datasets totaling 561k images generated by 12 GANs, 52 diffusion models, and 2 autoregressive models, including 3 in-the-wild datasets.** Generator types are indicated in parentheses (G = GAN, D = Diffusion, AR = Auto-Regressive). All detectors are evaluated using official checkpoints. To mitigate format bias, JPEG compression (quality 96) is applied to GenImage, ForenSynth, and AIGCDetectionBenchmark. "DDA (ours) + JPEG Aug" denotes training with additional random JPEG compression augmentation.
>
> | Method                    | GenImage | DRCT-2M  | DDA-COCO | EvalGEN  | Synthbuster | ForenSynth | AIGCDetection  Benchmark | Chameleon | Synthwildx | WildRF | Avg     | Min    |
> | --------------------------|----------|----------|----------|----------|-------------|------------|--------------|-----------|----------|----------|----------------|--------|
> |                           | 1G + 7D  | 16D      | 6D       | 5D + 2AR | 9D          | 11G        | 7G + 9D      | Unknown   | 3D       | Unknown  |                |        |
> | NPR (CVPR'24)             | 51.5     | 37.3     | 28.1     | 59.2     | 50.0        | 47.9       | 53.1         | 59.9      | 49.8     | 63.5     | 50.0 ± 10.7    | 28.1   |
> | UnivFD (CVPR'23)          | 64.1     | 61.8     | 3.6      | 15.4     | 67.8        | 77.7       | 72.5         | 50.7      | 52.3     | 55.3     | 52.1 ± 24.2    | 3.6    |
> | FatFormer (CVPR'24)       | 62.8     | 52.2     | 3.3      | 45.6     | 56.1        | *90.1*     | 85.0         | 51.2      | 52.1     | 58.9     | 55.7 ± 23.6    | 3.3    |
> | SAFE (KDD'25)             | 50.3     | 59.3     | 0.5      | 1.1      | 46.5        | 49.7       | 50.3         | 59.2      | 49.1     | 57.2     | 42.3 ± 22.3    | 0.5    |
> | C2P-CLIP (AAAI'25)        | 74.4     | 59.2     | 2.0      | 38.9     | 68.5        | **92.1**  | 81.4          | 51.1      | 57.1     | 59.6     | 58.4 ± 25.0    | 2.0    |
> | AIDE (ICLR'25)            | 61.2     | 64.6     | 1.2      | 15.0     | 53.9        | 59.4       | 63.6         | 63.1      | 48.8     | 58.4     | 48.9 ± 22.3    | 1.2    |
> | DRCT (ICML'24)            | 84.7     | 90.5     | 30.4     | 77.7     | 84.8        | 73.9       | 81.4         | 56.6      | 55.1     | 50.6     | 68.6 ± 19.4    | 30.4   |
> | AlignedForensics (ICLR'25)| 79.0     | 95.5     | 86.6     | 77.0     | 77.4        | 53.9       | 66.6         | 71.0      | 78.8     | 80.1     | 76.6 ± 11.2    | 53.9   |
> | **DDA (ours)**           | **95.5** | *97.4*   | **94.3** | *94.0*   | **94.6**   | 85.5       | **93.3**     | *74.3*    | *84.0*   | **95.1** | *90.8 ± 7.3*   | *74.3* |
> | **DDA (ours) + JPEG Aug**| *94.3*   | **97.9**| *92.8*    |**98.3** | *88.8*      | 83.2       | *89.6*       | **81.7**  | **88.0** | *94.9*   | **91.0 ± 5.7**|**81.7**|
>
> ---
>
> > *Q3: Address Performance on Heavily Post-Processed Images (Chameleon Dataset). Question: The paper acknowledges that "Our method performs relatively lower on FLUX... and struggles to detect images with strong post-processing artifacts, as shown in the results on the Chameleon dataset". Given that real-world scenarios often involve aggressive post-processing, how do the authors envision mitigating this limitation? Is DDA inherently sensitive to certain types of artifacts, or are there planned extensions to improve robustness in these challenging cases?*
>
> Thank you for highlighting this important concern. We address this limitation through two directions: (1) enhancing DDA's robsutness to heavily post-processing, and (2) integration DDA with vision-language models (VLMs) to incorporate semantic-level signals.
>
> **(1) Enhance DDA's robustness:** While DDA’s performance declines on challenging datasets like Chameleon, it still **outperforms all existing methods**. Moreover, as shown in the Table 1 (referenced in our response to Q2), applying stronger data augmentations (e.g., random JPEG compression) could effectively improve robustness, enabling DDA to further achieve **81% balanced accuracy on Chameleon**. To our knowledge, this marks the **first detector to exceed 80% accuracy**.
>
> **(2) Integration with VLM:** To further enhance resilience to aggressive edits, we plan to incorporate **semantic-level cues** that persist through low-level corruption—such as implausible object configurations (e.g., “a person with three hands”) or physically impossible scenes. These high-level inconsistencies complement DDA’s pixel- and frequency-level modeling. In preliminary experiments, prompting **Qwen2.5-VL-32B** as *“RealismNet, a multimodal expert who determines whether an image could be photographed in the real world without digital manipulation”* allowed the model to consistently flag semantically implausible content, suggesting strong potential as a complementary detection signal.
>
> Our future work will explore a **hybrid detection framework**, where a vision-language model helps localize reliable regions in the image that are less affected by post-processing. DDA can then focus on those regions to detect subtle generation artifacts. This synergy between **semantic robustness and low-level generalization** offers a promising path toward robust, real-world AI-generated image detection.
>
> ---
>
> > *Q4: "Re-evaluate "Theory Assumptions and Proofs" Claim for Clarity.*
>
> Thank you for pointing this out. We acknowledge the misunderstanding regarding the checklist item on "Theory Assumptions and Proofs." Section 3.2 provides methodological intuition rather than formal theorems or proofs. We will revise our response to “\[N/A]” to more accurately reflect the content of the paper.
>
> ---
>
> [1] Rethinking the Up-Sampling Operations in CNN-based Generative Network for Generalizable Deepfake Detection, CVPR 2024.
>
> [2] Spatial-Phase Shallow Learning: Rethinking Face Forgery Detection in Frequency Domain, CVPR 2021.

---

> ### Author Response · Authors · 2025-08-06
>
> **Dear Reviewer,**
>
> Thank you for your encouraging recognition of our work. We truly appreciate the time, effort, and thoughtful advice you have provided throughout the review process. **As the discussion period comes to an end, we look forward to your reflections on our responses and are happy to further clarify any points.**
>
> Best regards,
>
> Authors

---

### Comment · Area_Chair_kR8w · 2025-08-06
**[General Reminder for Authors and Reviewers] Author-Reviewer Discussion Phase Ending Soon**

Dear Authors and Reviewers,

As you know, the deadline for author-reviewer discussions has been extended to August 8. If you haven’t done so already, please ensure there are sufficient discussions for both the submission and the rebuttal.

Reviewers, please make sure you complete the mandatory acknowledgment **AND** respond to the authors’ rebuttal, as requested in the email from the program chairs.

Authors, if you feel that any results need to be discussed and clarified, please notify the reviewer. Be concise about the issue you want to discuss.

Your AC

---

### Note · Authors · 2025-08-11

We sincerely thank reviewers for their constructive comments and recognition of our work. (R1: xaYE, R2: sHpt, R3: ks8s, R4: Two9)

Our work, **Dual Data Alignment (DDA)**, adopts a **dataset-centric** approach for generalizable AIGI detection. By creating **closely aligned real–synthetic pairs in both pixel and frequency domains**, DDA enables models to learn more generalizable decision boundaries **directly from data**. We believe that **data is the most direct and effective means to make models truly generalizable**. Additionally, we propose **two new benchmarks** — the challenging **DDA-COCO** and the timely **EvalGEN** dataset.

Through the discussion period, we have addressed all the major concerns raised by the active reviewers (**R1, R3, R4 increased/maintained to score 5**). We also hope we have sufficiently addressed R2's concern, as **NO FURTHER CONCERN were raised after our response**. Below, we summarize the strengths acknowledged by reviewers and key responses to their concerns.

### Reviewer‑acknowledged strengths:

* **[Motivation & Novelty] (R1-R3):** Acknowledged for introducing **dual-domain alignment**, particularly addressing the **overlooked frequency-level bias**.
* **[Extensive Validation] (R1-R4):** **A single model trained exclusively on DDA-aligned COCO** is evaluated on **10 benchmarks** (**561k** images from 12 GANs, 52 diffusion, and 2 autoregressive models, plus **3 in-the-wild datasets**). DDA achieves SOTA on **9/10 benchmarks**, with the **highest average accuracy** (**↑14.2**), **lowest standard deviation** (**↓3.9**), and **highest worst-case accuracy** (**↑20.4**). To our knowledge this is the **most comprehensive benchmark evaluation**.

### Key responses to reviewers' concerns:

* **[Isolated Evaluation] (R2–R4):** Applying DDA to multiple baselines isolates its effect: UnivFD **↑28.2**, FatFormer **↑9.2**, DRCT **↑8.8** (*R2 Table 4*).
* **[JPEG Augmentation] (R2):** **DDA addresses format bias that JPEG compression alone does not**. When tested against JPEG compression, DDA remains robust (**↑3.0**) while VAE Rec.+JPEG Aug shows a significant drop (**↓22.0**). Furthermore, DDA is compatible with JPEG augmentation: on the in-the-wild dataset Chameleon, DDA + JPEG Aug achieves **81.7**, **the first detector above 80** (*R2 Table 1, 2*).
* **[Backbone & Input‑Size Ablations] (R2, R4):** Reported in the Appendix — DDA shows consistent superiority (**↑10.2**) with varied backbones and input sizes (*R4 Table 2, 3*).

---

### Decision · Program_Chairs · 2025-09-17

**Decision:**

Accept (spotlight)

**Comment:**

The recommendation is based on the reviewers' comments, the area chair's evaluation, and the author-reviewer discussion.

This paper proposes a dual data alignment (DDA) approach for both pixel and frequency domains. The resulting synthetically generated images are shown to be effective in improving the detection performance of various AI-generated image classifiers. All reviewers find the studied setting novel and the results provide new insights. The major concern of the initial version was on the fair evaluation versus baselines, given that the baseline detectors were trained on different datasets. The authors’ rebuttal has successfully addressed the major concerns of reviewers, by providing a controlled experiment (one-to-one comparison) on baseline models with and without DDA.

In the post-rebuttal phase, most reviewers were satisfied with the authors’ responses and agreed on the decision of acceptance. Overall, I recommend acceptance of this submission. I also expect the authors to include the new results and suggested changes during the rebuttal phase in the final version.

Also, given that the proposed method is quite generic and improves different AIGI detectors at large with notable gains, I recommend for spotlight presentation.